# Dissecting muscle synergies in the task space

**David O'Reilly\*, Ioannis Delis\***

School of Biomedical Sciences, University of Leeds, Leeds, United Kingdom

**Abstract** The muscle synergy is a guiding concept in motor control research that relies on the general notion of muscles '*working together*' towards task performance. However, although the synergy concept has provided valuable insights into motor coordination, muscle interactions have not been fully characterised with respect to task performance. Here, we address this research gap by proposing a novel perspective to the muscle synergy that assigns specific functional roles to muscle couplings by characterising their task-relevance. Our novel perspective provides nuance to the muscle synergy concept, demonstrating how muscular interactions can '*work together*' in different ways: (1) irrespective of the task at hand but also (2) redundantly or (3) complementarily towards common task-goals. To establish this perspective, we leverage information- and network-theory and dimensionality reduction methods to include discrete and continuous task parameters directly during muscle synergy extraction. Specifically, we introduce co-information as a measure of the task-relevance of muscle interactions and use it to categorise such interactions as task-irrelevant (present across tasks), redundant (shared task information), or synergistic (different task information). To demonstrate these types of interactions in real data, we firstly apply the framework in a simple way, revealing its added functional and physiological relevance with respect to current approaches. We then apply the framework to large-scale datasets and extract generalizable and scale-invariant representations consisting of subnetworks of synchronised muscle couplings and distinct temporal patterns. The representations effectively capture the functional interplay between task end-goals and biomechanical affordances and the concurrent processing of functionally similar and complementary task information. The proposed framework unifies the capabilities of current approaches in capturing distinct motor features while providing novel insights and research opportunities through a nuanced perspective to the muscle synergy.

**\*For correspondence:**
bsdor@leeds.ac.uk (DO'R);
i.delis@leeds.ac.uk (ID)

**Competing interest:** The authors declare that no competing interests exist.

## eLife assessment

The work by O'Reilly and Delis is **important** to extend the synergy ideas using methods from signal processing and information theory to cluster muscles and task parameters, thereby advancing our understanding of the modular architecture of motor control. The method is innovative, and the findings are **compelling** from theoretical and practical perspectives. The work will be of broad interest to motor control and neural engineering researchers.

## Introduction

Human movement is a highly complex behaviour, with a broad spectrum of multiplexed spatio-temporal dynamics typically exhibited for basic activities-of-daily-living (*Macpherson et al., 2021*; *Kaplan et al., 2020*). How the central nervous system controls movement in the face of this inherent complexity to ensure efficient and reliable navigation of the environment and task performance is a nontrivial question currently under investigation in the motor control field (*Bruton and O'Dwyer, 2018*; *Bizzi and Cheung, 2013*). The muscle synergy hypothesis is a long-withstanding proposition on the underlying neural constraints producing coordinated movement, stating that this complexity is

offset by the allocation of computational resources to the spinal level in the form of motor primitives (**Bruton and O'Dwyer, 2018**; **Bizzi and Cheung, 2013**; **Berret et al., 2019**; **Bernstein, 1966**). These motor primitives modularly activate functional groups of muscles that are flexibly combined for the efficient construction of a movement. The conceptual underpinning of the '*muscle synergy*' entails the idea of combinations of muscles '*working together*' for the purpose of effective goal-orientated behaviour (**Latash, 2008**). This emergent cohesion involves the following qualifying attributes: a repeatable muscle activation pattern common across trials and participants, a reciprocal relationship among functional muscle groups such that changes occur in one group to compensate for changes in another, and task dependence (i.e. the pattern of interdependencies among muscles must map onto task performance) (**Latash, 2008**; **Brenner et al., 2000**). A common approach to analyse the neural constraints underlying these motor patterns is to apply unsupervised machine-learning algorithms to electromyographic (EMG) data (**Turpin et al., 2021**; **d'Avella and Lacquaniti, 2013**), with the aim of extracting a latent, low-dimensional representation.

In **Ó' Reilly and Delis, 2022**, we considered, key limitations among current approaches to muscle synergy analysis in extracting functionally relevant and interpretable patterns of muscle activity (**Alessandro et al., 2013**). We proposed a combinatorial approach based on information- and network-theory and dimensionality reduction (the network information framework [NIF]) that significantly improved the generalisability of the extraction process by, among others, removing restrictive model assumptions (e.g. linearity, same mixing coefficients) and the reliance on variance-accounted-for metrics (**Alessandro et al., 2013**). By determining the pairwise mutual information (MI) between muscles, this innovation paved the way for the appropriate mapping of muscular interactions to the task space. To elaborate on the significance of this development, the extraction of motor patterns in isolation of the task space often comes at the expense of functional and physiological relevance (**Alessandro et al., 2013**; **de Rugy et al., 2013**). Furthermore, effective methods for mapping large-scale physiological dynamics to behaviour is a current gap across the neurosciences (**Krakauer et al., 2017**). Thus, here we build on this work by, for the first time, directly including task space parameters during muscle synergy extraction. This enables us, in a novel way, to dissect the concept of the muscle synergy and therefore quantify interactions between muscle activations with shared or complementary functional roles.

Further to the above, in its currently defined state, the muscle synergy concept describes the role of common neural drives to functional muscle groupings working redundantly towards a common task-goal (**Bruton and O'Dwyer, 2018**; **Bizzi and Cheung, 2013**; **Berret et al., 2019**; **Latash, 2008**; **Cheung and Seki, 2021**). However, recent influential works have highlighted several other important mechanisms involved in this low-dimensional control strategy that are not well recognised by the muscle synergy concept (**Ronzano et al., 2021**; **Hug et al., 2021**; **Hug et al., 2023**; **Alessandro et al., 2020**; **Valero-Cuevas et al., 2009**; **Nazarpour et al., 2012**; **Todorov and Jordan, 2002**; **d'Avella and Bizzi, 2005**; **Cheung et al., 2005**). Such insights include the partitioning of motor variability by the nervous system into task-relevant and -irrelevant spaces and the cooperation between functionally distinct muscle groupings in the form of cross-module functional connectivities. These observations highlight the need for a refinement of the muscle synergy concept to comprehensively describe diverse muscle interactions during movement, including their partitioning into task-relevant and -irrelevant spaces and the characterisation of their functional roles.

We thus motivate the development of a more nuanced perspective to the muscle synergy concept and the general notion of '*working together*' that comprehensively describes the muscle interactions underlying motor behaviour. To do so, we propose an information-theoretic approach (based on the NIF pipeline) that characterises the contributions of muscle couplings to task performance. In other words, we frame the notion of '*working together*' in more specific terms of shared information between pairs of spatiotemporal muscle activations ($[m_x, m_y]$ **Figure 1.3a**) (red and green sets, respectively) and a corresponding task parameter ($\tau$) (blue set) (see Venn diagrams in **Figure 1.3b**). Among current approaches to muscle synergy analysis, the shared information between muscles (yellow and white areas in **Figure 1.1a**) is quantified, using dimensionality reduction, as common patterns of variability. These common patterns are essentially task-agnostic and may contain patterns of variability (1) present in specific tasks (i.e. task-relevant white shaded area in **Figure 1.1a** as well as 2) shared across tasks (i.e. task-irrelevant, yellow shaded area in **Figure 1.1a**). Our proposed approach dissects patterns of muscle variability in space, time, and across trials in terms of their task-relevance and

**Figure 1.** A general outline of the proposed approach. (**1.1a, b**) We propose a novel approach to mapping muscle couplings to the task space. Among current muscle synergy analysis approaches, muscle couplings are quantified in isolation of the task solely using dimensionality reduction. Using our approach, the functional characteristics of muscle interactions can be quantified in terms of the similarity of their encoded task information. We do so by determining the coupling between $[m_x, m_y]$ and a corresponding task parameter ($\tau$) using mutual information (MI). From this perspective, task-redundant muscle couplings (pink shaded area in pink-orange intersection) represent muscles cooperating towards similar task-goals, while task-synergistic muscle couplings (orange shaded area in pink-orange intersection) encapsulate the task information provided by a muscle pairing acting towards complementary task-goals. Muscle couplings present across tasks (i.e. task-irrelevant) are quantified by conditioning the MI between $[m_x, m_y]$ pairs with respect to $\tau$ (yellow intersection). (**1.2**) A description of redundant and synergistic interactions. (**a**) Net redundant interactions are defined by a greater amount of information generated by the sum of individual observation of $m_x$ and $m_y$ ($[m_x + m_y]$) than their simultaneous observation ($[m_x, m_y]$). (**b**) In a net synergistic interaction, $[m_x, m_y]$ provides more information than $[m_x + m_y]$. (**1.3a, c**) An overview of the approach. Spatiotemporal muscle

*Figure 1 continued on next page*

*Figure 1 continued*

activation samples are extracted across trials from large-scale electromyographic (EMG) datasets and concatenated into vectors, forming $[m_x, m_y]$ pairs. The derived muscle couplings are then run through the network information framework (NIF) pipeline (*Ó' Reilly and Delis, 2022*), producing low-dimensional, multiplexed space–time muscle networks.

functional similarity in a generalizable manner using MI (*Figure 1.3a–c*). This enables us to decompose muscle activations into muscle pair–task parameter couplings and characterise their combined functional roles. We can then extract low-dimensional representations of these muscle couplings, that is muscle networks with specific spatial and temporal signatures, across participants and tasks (*Figure 1.3c*).

Crucially, using this novel framework, we can separately quantify the task-irrelevant (i.e. muscle interactions present across tasks) information conveyed by a muscle coupling (yellow intersection in *Figure 1.1b*) from the task-relevant information (pink-and-orange shaded area in *Figure 1.1b*). These task-relevant interactions can be either sub-additive/redundant (i.e. the muscle coupling provides less information about the task compared to the sum of the individual muscle patterns) or super-additive/synergistic (i.e. the muscle coupling conveys more task information than the sum of individual muscle-task encodings). Conceptually, the information a muscle interaction provides is considered redundant when all the information can essentially be found in one of the muscles (pink shaded area *Figure 1.2a*). This redundant task information thus reveals a functional similarity between muscle activations. Alternatively, we can also identify muscles that act synergistically towards complementary task-goals, meaning their variations provide different information about a motor behaviour. A key, quantifiable attribute of this complementary interaction is the emergent task information (*synergy*) they provide when considered together (orange shaded area *Figure 1.2b*). From this novel perspective, muscle activations can '*work together*' not just similarly towards a common task-goal but also complementarily towards different aspects of motor behaviour and concurrently towards objectives functionally irrelevant to overt task performance, thus providing a comprehensive view of the muscle interactions governing coordinated movement.

To illustrate this novel conceptual and analytical framework, we conducted two example applications to data from human participants performing naturalistic movements. These applications demonstrate the added utility of this framework to current muscle synergy analysis in terms of functional and physiological relevance and interpretability. We then applied it to three large-scale datasets, extracting generalizable and functionally interpretable space–time muscle networks with respect to

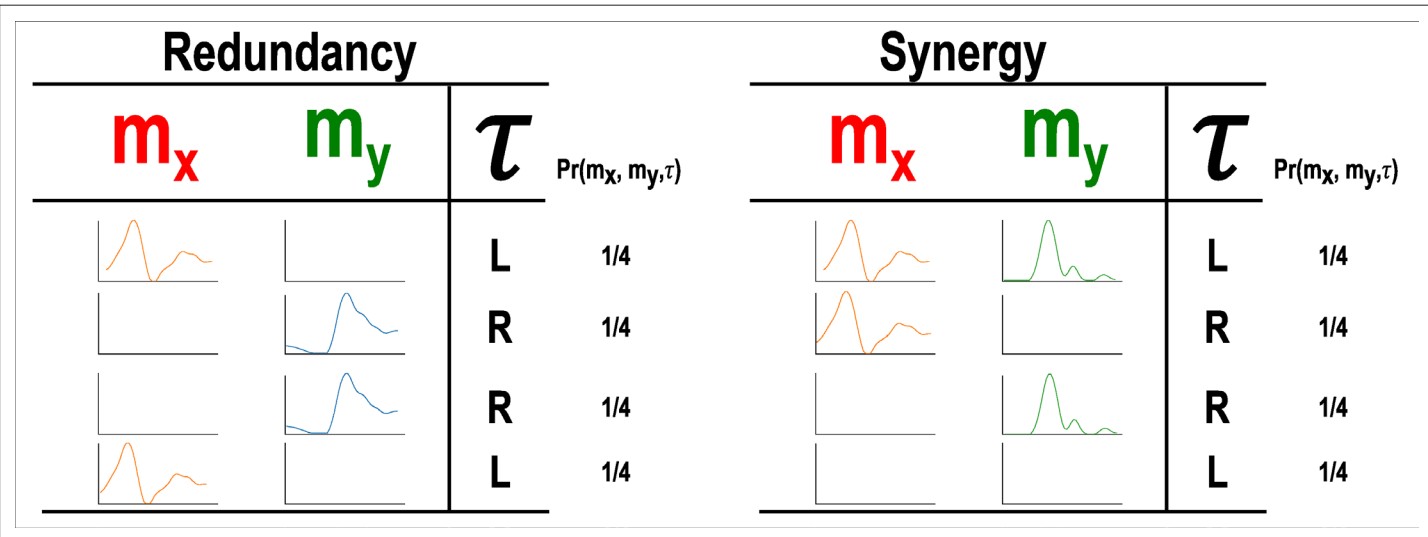

**Figure 2.** A simulation demonstrating how informational redundancy and synergy can be interpreted when applied to the muscle space. Four observations of a given muscle pair ($m_x$ and $m_y$) that can fall into two equiprobable on and off activation states and a corresponding task parameter ($\tau$) describing left (L) or right (R) movement direction. Observing either $m_x$ or $m_y$ in the redundancy example gives 1 bit of information while observing both $m_x$ and $m_y$ together in the synergy example gives 1 bit of information.

both discrete and continuous task spaces. We have also made available open-source Matlab routines for readers to apply this approach to their own data (https://github.com/DelisLab/EMG2Task copy archived at *O'Reilly and Delis, 2024*).

## Results

Our primary aim here is to characterise muscle synergies in task space by quantifying the contributions of muscle couplings to task performance. To achieve this, we essentially reverse the analytical approach typically used in muscle synergy studies (i.e. muscle groupings are identified and inferences then made about their functional roles) (*Turpin et al., 2021*). More specifically, we firstly identify functional couplings between paired muscle activations by evaluating their task-relevance and then extracting representative patterns of such couplings using dimensionality reduction methods. This enables us to distinguish task-irrelevant from task-relevant muscle couplings. Of the muscle couplings that demonstrate task-relevance, we can then characterise their functional roles as either redundant or synergistic. *Figures 2 and 3* illustrates a simulation to facilitate interpretations of what informational redundancy and synergy mean when applied to muscle activities in the context of task performance. Redundant task information is generated when $m_x$ and $m_y$ carry identical predictive information about $\tau$. This is distinct from current muscle synergy analysis which would consider $m_x$ and $m_y$ to share information about $\tau$ if their magnitudes are equivalent. Here, $\tau$ is always L when $m_x$ is on and when $m_y$ is off and R when vice versa. Thus, all the task information can be found in either $m_x$ or $m_y$ alone, generating 1 bit of redundant information. Synergistic task information on the other hand is predictive task information generated only when observing both $m_x$ and $m_y$ together. In the simple example shown, $\tau$ can be L when both $m_x$ and $m_y$ are active or inactive. However, we can see that when both muscles are active or inactive then $\tau$ is L. Thus, no predictive task information is provided by either $m_x$ or $m_y$ alone but the full 1 bit of information available is generated when observing both muscles together .

### Building on current approaches to muscle synergy analysis

Current approaches to muscle synergy analysis based on non-negative matrix factorisation (NMF) have a proven use case in the extraction of functionally and physiologically relevant motor patterns (*Bruton and O'Dwyer, 2018*; *Alessandro et al., 2013*; *Funato et al., 2022*; *Scano et al., 2022*; *Buongiorno et al., 2020*). To demonstrate that the proposed framework adds to this current utility, here we firstly provide a simple example output from the proposed and current approaches (see Figure 5A–D). This example was derived from the EMG recordings of a single trial of a participant walking on level ground in the counter-clockwise direction around the circuit depicted in *Figure 4C*, *Camargo et al., 2021*. For the proposed approach, the muscle couplings were determined with respect to a single, continuous task parameter, the heel kinematic marker in the anterior–posterior direction. For the application of the current approach, we applied the spatial muscle synergy model across the same single-trial EMG recordings (*Tresch et al., 1999*), extracting one component.

The simplified representations from the proposed approach reveal the functional role of muscular interactions with respect to the heel marker and provide intuition on the types of muscle couplings that can be identified, including task-irrelevant (A), -redundant (B), and -synergistic (C) interactions. Their submodular structure is illustrated via the node colour on the accompanying human body models (*Makarov et al., 2015*), describing muscles that have a closer functional relationship. For example, (1) the hamstring muscles (ST and BF) controlling knee flexion work redundantly together with muscles involved in hip abduction (GR, GlutM, and Obl) to move the heel around the circuit (*Figure 4B*), and (2) calf muscles involved in ankle flexion (GM and TA) cooperatively determine heel position in synergy with the same hamstring muscles (*Figure 5C*Figure 5C). Moreover, these hamstring muscles together with their antagonist RF also form a task-irrelevant network, that is their couplings are not predictive of heel position (*Figure 5A*). Overall, task-irrelevant muscle couplings primarily capture interactions between co-agonist and agonist–antagonist muscles that are indiscriminative of heel marker position. Also, task-redundant and -synergistic couplings reveal functionally similar and dissimilar muscle combinations, respectively, that provide sub-additive (i.e. shared or redundant) and super-additive (i.e. complementary or synergistic) information about heel marker position.

The NMF representation (*Figure 5D*) conveys important information about gait in a task-agnostic manner, thus it may contain both task-relevant and -irrelevant interactions. Intuitively, SO, GM, and

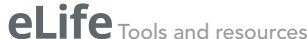

**Figure 3.** A summary of the network information framework (NIF) pipeline. (**A**) Large-scale datasets of electromyographic (EMG) signals are captured while participants perform various motor tasks (*Delis et al., 2014*; *Hilt et al., 2018*; *Camargo et al., 2021*). (**B**) The mutual information (MI) between all unique muscle-timepoint vector ($[m_x, m_y]$ combinations with respect to a corresponding task parameter ($\tau$) is determined (*Ince et al., 2017*), forming a network of functional connectivities. (**C**) These adjacency matrices are then analysed in terms of statistical significance and modular structure using

*Figure 3 continued on next page*

*Figure 3 continued*

percolation theory (*Ó' Reilly and Delis, 2022*). (**D**) The optimal spatial and temporal model-ranks are determined using generalised, consensus-based network community detection methods (*Blondel et al., 2008*; *Mucha et al., 2010*; *Lancichinetti and Fortunato, 2012*; *Rubinov and Sporns, 2010*). (**E**) The optimal model-ranks are used as input parameters for dimensionality reduction, where space–time muscle networks along with their underlying activation coefficients are concurrently extracted (*Delis et al., 2014*).

**Figure 4.** Graphical illustrations of each of the datasets analysed in the current study. (**A**) Dataset 1 consisted of participants executing table-top point-to-point reaching movements (40 cm distance from starting point P0) across four targets in forward (P1–P4) and backwards (P5–P8) directions at both fast and slow speeds (40 repetitions per task) (*Delis et al., 2014*). The muscles recorded included the finger extensors (FE), brachioradialis (BR), biceps brachii (BI), medial-triceps (TM), lateral-triceps (TL), anterior deltoid (AD), posterior deltoid (PD), pectoralis major (PE), latissimus dorsi (LD) of the right, reaching arm. (**B**) For dataset 2, the activity of 30 muscles was recorded while participants performed whole-body point-to-point reaching movements across three different heights and bars and in various directions, accumulating to 72 unique reaching tasks (*Hilt et al., 2018*). (**C**) The circuit navigated by participants in dataset 3 as they executed various locomotion modes is illustrated, of which level-ground walking, stair- and ramp-ascent/descent were analysed in the current study (*Camargo et al., 2021*). Several sub-conditions were undertaken by participants for each locomotion mode including different walking speeds, clockwise vs. counter-clockwise direction, different stair heights and ramp inclines, etc. Participants executed these tasks while the electromyography (EMG) of 11 muscles on the right leg gluteus medius (GlutM), right external oblique (Obl), semitendinosus (ST), gracilis (GR), biceps femoris (BF), rectus femoris (RF), vastus lateralis (VL), vastus medialis (VM), soleus (SO), tibialis anterior (TA), gastrocnemius medialis (GM) along with kinematic, dynamic, and inertial motion unit (IMU) signals were captured. (**D**) The EMG placement for dataset 4 [deltoideus pars clavicularis (DC), biceps brachii (BB), triceps brachii (TB), flexor digitorum superficialis (FDS), extensor digitorum (ED), brachioradialis (BR), flexor carpi ulnaris (FCU), extensor carpi ulnaris (ECU), pronator teres (PT), flexor carpi radialis (FCR), abductor pollicus brevis (APB), abductor digiti minimi (ADM)] (*Averta et al., 2021*). A single trial was taken from 25 healthy and 20 post-stroke participants performing a unilateral pointing movement with the index finger and arm outstretched (task 9 of the Softpro protocol [MHH]).

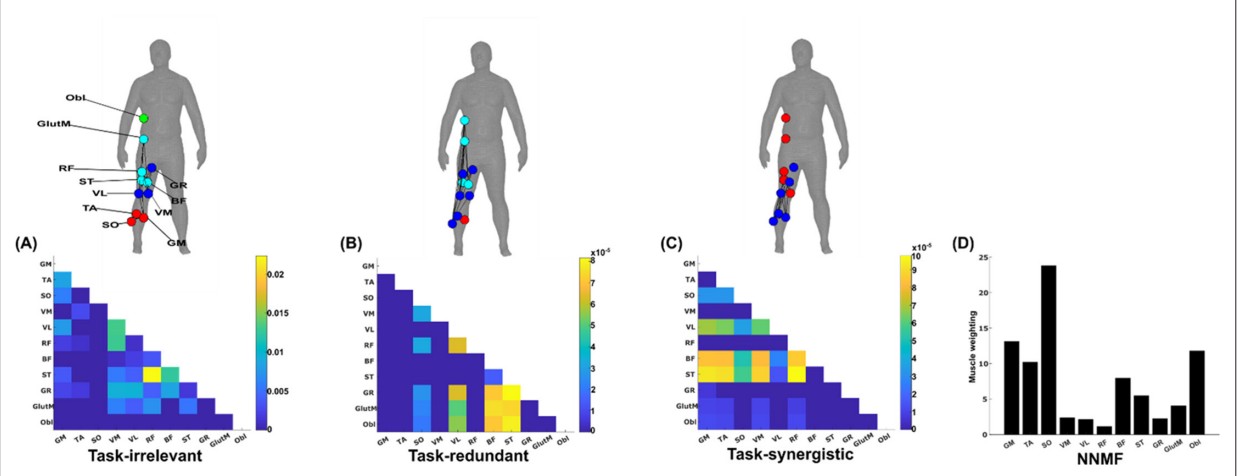

**Figure 5.** A simplified example output from the proposed framework applied to a single trial of turning gait from dataset 3. (**A**) Task-irrelevant, (**B**) task-redundant, and (**C**) task-synergistic synchronous muscle couplings were quantified (the unit of shared information is 1 bit) with respect to the heel kinematic marker (anterior–posterior direction). Human body models accompanying each spatial network illustrate their respective submodular structure with node colour and size and edge width indicating community affiliation (**Blondel et al., 2008**), network centrality and connection strength, respectively (**Makarov et al., 2015**; **Benzi and Klymko, 2013**). (**D**) A corresponding synergy representation from a single trial of turning gait from dataset 3 extracted using the spatial model from current approaches (**Tresch et al., 1999**). Each bar represents the relative weighting of each muscle in the synergy component.

Obl of the outer, right side are most prominently weighted here, perhaps representing their functional role in accelerating the body in the counter-clockwise direction around the circuit whilst maintaining upright posture (**Figure 4C**). This observation can only be inferred indirectly however as, amongst current approaches, no direct association is made with other muscles or with task performance. Indeed, with respect to heel position, the proposed approach reveals that SO is functionally similar to Obl and GM whilst also containing task-synergistic and -irrelevant information with GM (**Figure 5A–C**). From the perspective of existing approaches, the knee flexors and extensors play a minor role during turning gait (as indicated by their relatively low weighting **Figure 5D**). However, the proposed approach conveys a central role for these muscles suggesting that the proposed framework enables targeted dissections of muscle functionality with respect to a chosen task parameter, thus revealing subtle couplings with potentially important behavioural consequences.

Next, to demonstrate the additional physiological relevance the proposed methodology brings to current muscle synergy analysis, we applied the proposed framework to single trials of pointing movements performed by 20 participants with stroke and 25 healthy controls from dataset 4 (**Figure 4D**) ('MMH' task 9, **Averta et al., 2021**). Specifically, we determined MI from a randomly selected trial of healthy and post-stroke participants with respect to the 3D position of the anterior wrist kinematic marker (WRBA) of the pointing arm. We chose WRBA as the task variable here due to its sensitivity to hand orientation. For the purpose of this simplified demonstration, we focused on a comparison between task-redundant (with respect to WRBA) and NMF-based muscle representations. We firstly generated a normative representation of this pointing motion by extracting the first component across the healthy controls using the proposed approach (**Figure 6a.1** and NMF, **Figure 6a.2**). We then quantified the similarity of muscle representations extracted from each post-stroke participant individually to this normative reference (obtained across all healthy participants) using Pearson's correlation and converted these values to distances (i.e. 1-r) (**Figure 6b, c**). Finally, we determined if these distances from healthy control values were predictive of the stroke survivor's motor impairment, measured using the upper-extremity section of the Fugl-Meyer assessment (**Figure 6d**).

To briefly summarise the results, the distance of post-stroke participants from healthy controls was found to be predictive of motor impairment for the proposed approach ($\beta = -8.52 \pm 2.2$, p = 0.0012) but not the NMF-based approach ($\beta = 1.4 \pm 1.39$, p = 0.33). This finding suggests, intuitively, that the proposed approach captures redundant muscle couplings that support robust motor control and that deviations from this normative pattern of motor redundancy are linearly related to the degree of impairment. Importantly, this result was obtained using only one randomly selected trial for each

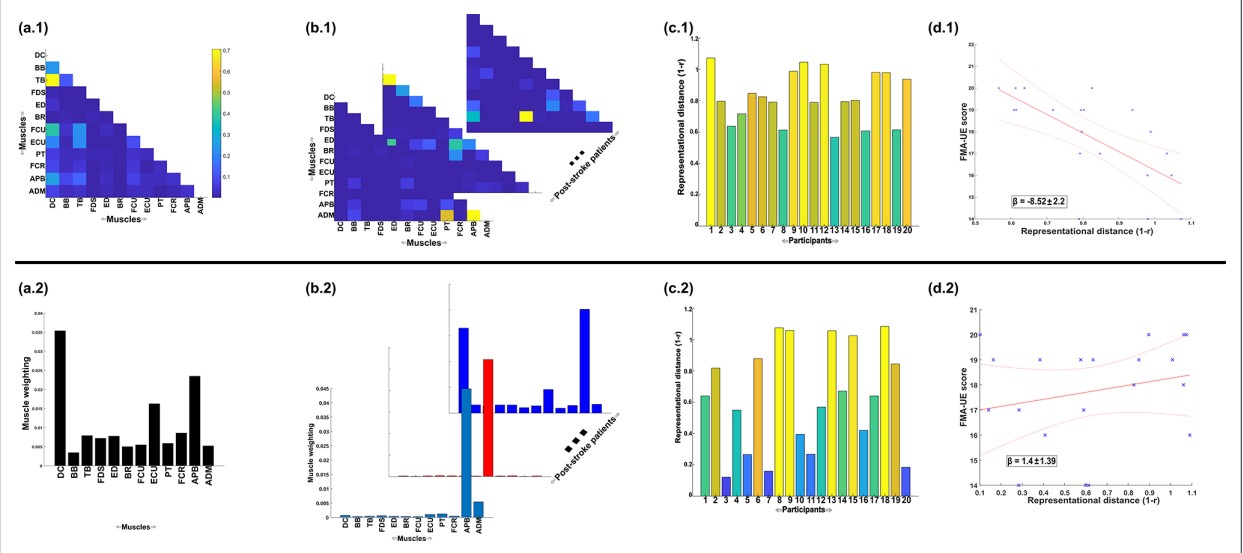

**Figure 6.** A simple demonstration of the physiological relevance of the proposed approach (a.1–d.1) and the traditional, non-negative matrix factorisation (NMF)-based approach (a.2–d.2). From dataset 4 (*Averta et al., 2021*), we took the electromyographic (EMG) signals and WRBA kinematic from 20 post-stroke and 25 healthy participants. We extracted a single normative reference of healthy controls task-redundant muscle couplings with respect to WRBA (a.1) and a corresponding normative reference using NMF only (a.2). We then extracted a single component from each post-stroke participant and compared them individually with the corresponding normative reference, computing distance values (1 r) (**b, c**). We finally determined the predictive relationship of these distance values with a measure of upper-extremity motor impairment derived from the Fugl-Meyer assessment (FMA-UE) (**d**).

participant. This simple example conveys the physiologically relevant targeted insights that can be generated from the proposed framework. Although current approaches have demonstrated significant linear relationships with motor impairment (*Clark et al., 2010*; *Schwartz et al., 2016*; *Steele et al., 2015*), these assessments generally rely on large numbers of trials and participants and do not point to specific underlying muscle interactions as provided here.

Together, through two example applications, we have demonstrated attributes of the proposed methodology that provide novel capabilities to current muscle synergy analysis. In the following, we sequentially present in more detail the three types of muscular interactions in the spatiotemporal domain across three datasets and then show the robustness of the approach and its outputs.

## Representations of motor behaviour in muscle couplings

To begin, we derived pairs of muscle activation vectors $[m_x, m_y]$ from benchmark datasets of human participants executing naturalistic movements, namely arm reaching (dataset 1), whole-body point-to-point reaching (dataset 2), and various locomotion modes (dataset 3) (*Delis et al., 2014*; *Hilt et al., 2018*; *Camargo et al., 2021*; *Averta et al., 2021*) (see *Figure 4* for the experimental design of each dataset and 'Materials and methods' for an outline of the experimental setups and EMG data pre-processing) (*Figure 3A*). For datasets 1 and 2, we determine the MI between $[m_x, m_y]$ vectors with respect to several discrete task parameters representing specific task attributes (e.g. reaching direction, speed, etc.), while for dataset 3 we determined the task-relevant and -irrelevant muscles couplings in an assumption-free way by quantifying them with respect to all available kinematic, dynamic, and inertial motion unit (IMU) features.

Having extracted the muscle pair-task interdependencies representing a specific intersection in *Figure 1.1b*, we next sought to find a parsimonious representation of motor behaviour that is consistent across tasks and participants for datasets 1–3 (*Figure 3E*; *Ó' Reilly and Delis, 2022*; *Delis et al., 2014*). To produce this sparse, low-dimensional representation, we undertook the following intermediary steps:

We modelled the MI values as adjacency matrices in the spatial or temporal domain and identified dependencies that were statistically significant using percolation theory (*Figure 3C*; *Gallos et al., 2012*; *Bunde and Havlin, 2012*). By assuming the muscle networks operate near a state of

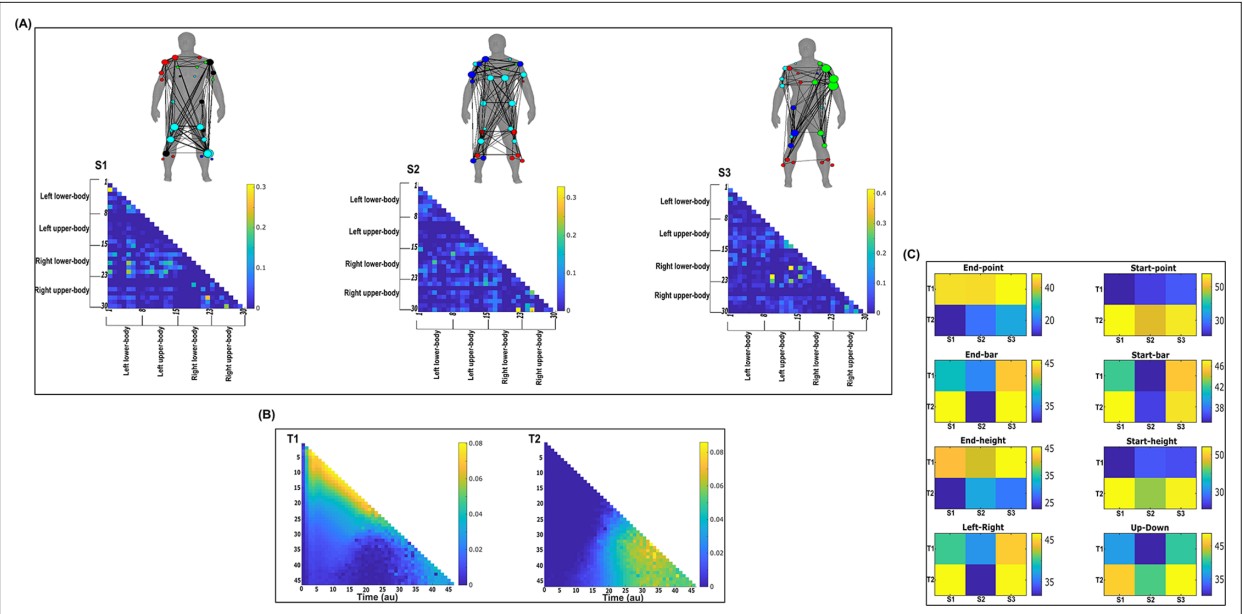

**Figure 7.** Three spatial (S1–S3) and two temporal task-irrelevant muscle networks (T1–T2) were empirically identified and extracted across participants and task parameters from dataset 2 using the network information framework (NIF) pipeline (panels A, B) (*Ó' Reilly and Delis, 2022*; *Hilt et al., 2018*). (Panel **C**) Activation coefficients are presented to the right of the networks, indicating their task parameter-specific scaling averaged across participants. Human body models accompanying each spatial network illustrate their respective submodular structure with node colour and size and edge width indicating community affiliation (*Blondel et al., 2008*), network centrality and connection strength, respectively (*Makarov et al., 2015*; *Benzi and Klymko, 2013*).

The online version of this article includes the following figure supplement(s) for figure 7:

**Figure supplement 1.** Three spatial (S1–S3) and two temporal task-irrelevant muscle networks (T1–T2) were empirically identified and extracted across participants and task parameters from dataset 1 using the network information framework (NIF) pipeline (panels **A, B**) (*Ó' Reilly and Delis, 2022*; *Delis et al., 2014*).

self-organised criticality (*Bak et al., 1987*), we effectively isolated dependencies that were above chance-level occurrence, thus empirically sparsifying the networks.

To empirically determine the number of components to extract in a parameter-free way, we then concatenated these adjacency matrices into a multiplex network and employed network community detection protocols to identify modules across spatial and temporal scales (*Figure 3D*; *Blondel et al., 2008*; *Mucha et al., 2010*; *Lancichinetti and Fortunato, 2012*; *Rubinov and Sporns, 2010*; *Didier et al., 2018*). Having detected the spatiotemporal modular structure, we then returned the sparsified networks to their original format and used the number of modules identified as input parameters into dimensionality reduction (*Figure 3E*; *Delis et al., 2014*).

By optimising a modularity-maximising cost-function (*Newman and Girvan, 2004*; *Magnani et al., 2022*), the community detection protocols we employed consistently identified three spatial (S1–S3) and two temporal (T1–T2) modules as representative of the underlying task-redundant, -synergistic, and -irrelevant informational dynamics. Following their extraction, we further analysed the spatial networks from each dataset in terms of their submodular structure by applying network-theoretic tools (*Blondel et al., 2008*; *Rubinov and Sporns, 2010*; *Benzi and Klymko, 2013*). In doing so, we identified subnetworks within each spatial network and interesting patterns of network centrality, that is the relative importance of a node in a network. The spatial and temporal networks of each dataset output are illustrated in panels A and B (*Figure 7*; *Figure 8*; *Figure 9*; *Figure 10*; *Figure 11*; *Figure 12*) of the following sections. They are accompanied by human body models where node colour and size indicate subnetwork community affiliation and network centrality, respectively (*Makarov et al., 2015*). The networks we extracted operate in parallel within spatial and temporal domains while having an all-to-all correspondence across domains, that is any spatial component can be combined with any temporal component via a task-specific coefficient (illustrated in panel C for dataset 1 and 2 outputs and in the supplementary materials for dataset 3) (*Delis et al., 2014*). Unlike similar muscle synergy

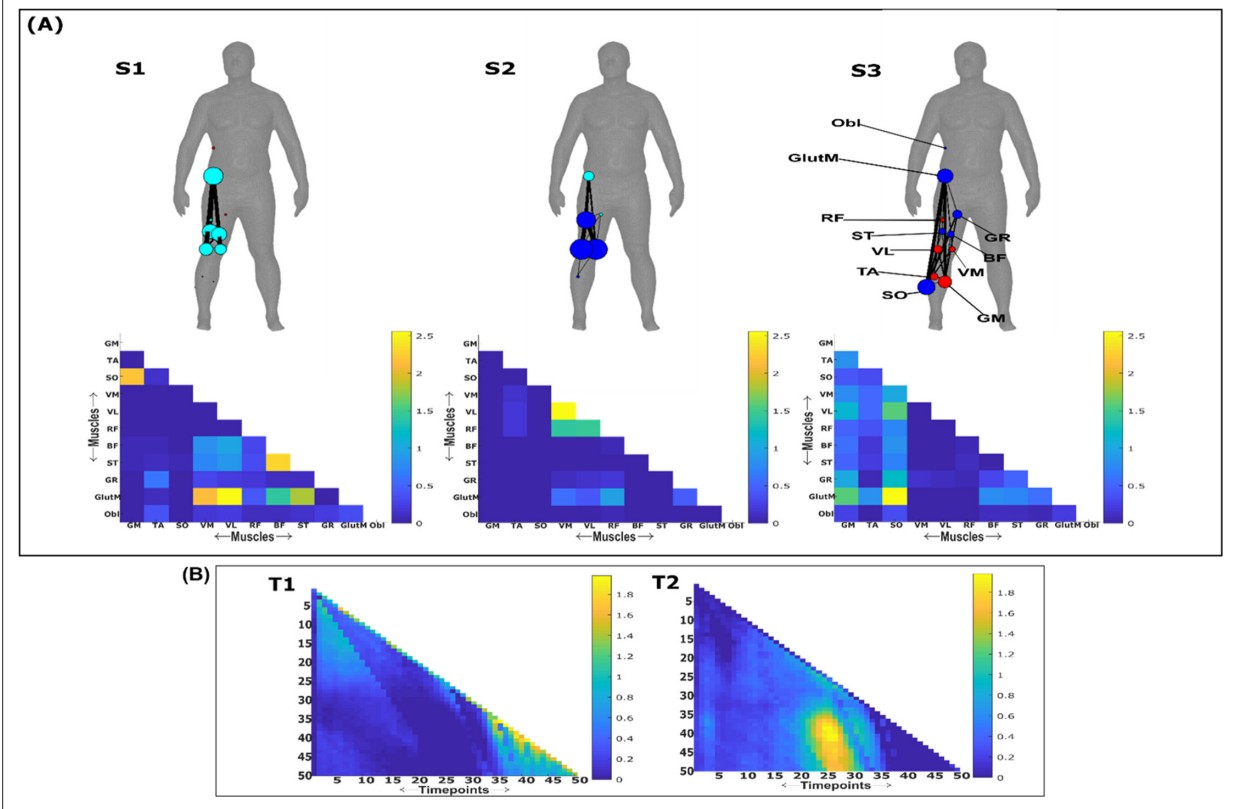

**Figure 8.** Three spatial (S1–S3) and two temporal task-irrelevant muscle networks (T1–T2) were empirically identified and extracted across participants and task parameters from dataset 3 using the network information framework (NIF) pipeline (panels **A, B**) (*Ó' Reilly and Delis, 2022*; *Camargo et al., 2021*). Activation coefficients are presented in supplementary material 1, indicating their task parameter-specific scaling averaged across participants in the dynamic, inertial motion unit (IMU), and kinematic spaces. Human body models accompanying each spatial network illustrate their respective submodular structure with node colour and size and edge width indicating community affiliation (*Blondel et al., 2008*), network centrality and connection strength, respectively (*Makarov et al., 2015*; *Benzi and Klymko, 2013*).

The online version of this article includes the following figure supplement(s) for figure 8:

**Figure supplement 1.** Task-irrelevant activation coefficients (dataset 3) (*Delis et al., 2014*).

extraction approaches, dimensionality reduction in the NIF pipeline does not seek to approximate the variance of recorded EMG data but to identify sets of muscles that share the same type of interaction. Thus, the multiplexing coefficients extracted in this framework are instead interpreted as the participant- and task-specific scaling of information overlap.

## Task-irrelevant muscle couplings

To quantify the task-irrelevant contributions of muscular interactions to motor behaviour, we conditioned the MI between $[m_x, m_y]$ with respect to $\tau$ (see 'Materials and methods') (*Ince et al., 2017*). This conditioning effectively removes the task-relevant information, leaving information produced by pairwise muscle variations that are task-indiscriminative. Following a run through the NIF pipeline (*Figure 3*; *Ó' Reilly and Delis, 2022*), the output from datasets 2 and 3 are presented in *Figures 7 and 8*, respectively, while the output from dataset 1 is presented in *Figure 7—figure supplement 1*. The task-irrelevant space–time muscle networks we extracted from datasets 1 and 2 shared several structural features with their task-agnostic counterparts extracted in the preliminary study (*Ó' Reilly and Delis, 2022*), supporting recent work showing that functional muscle network structure are heavily influenced by task-irrelevant factors such as anatomical constraints (*Kerkman et al., 2018*). The functional connectivities identified here captured known contributions of spatiotemporal muscular interactions to aspects of motor behaviour common across tasks and participants which we outline briefly here.

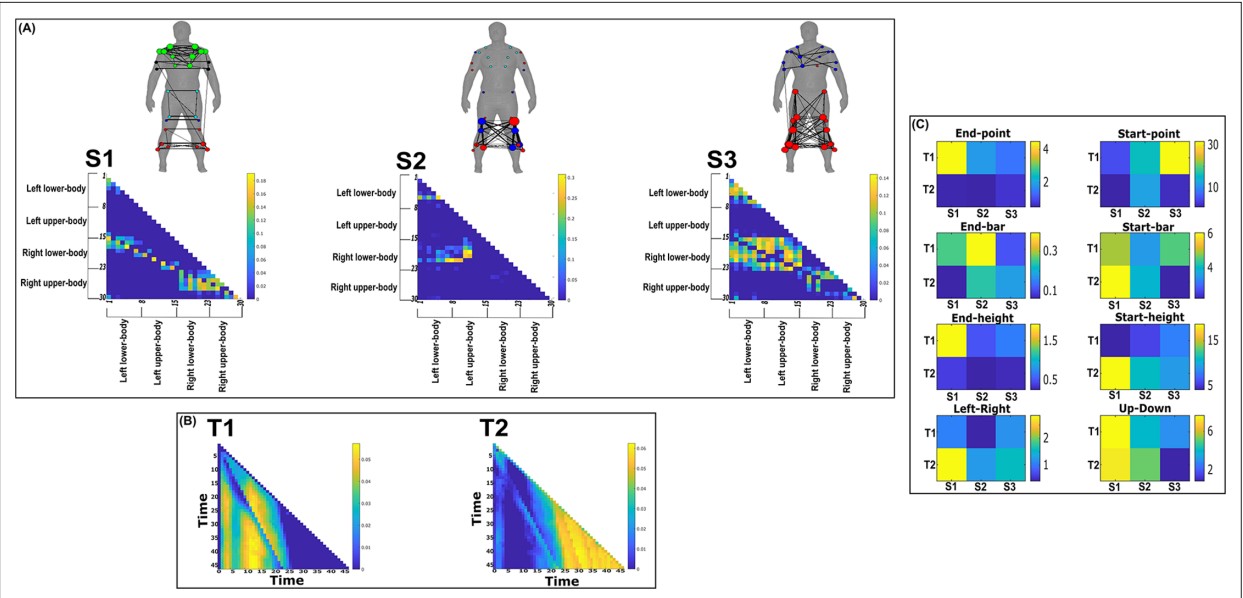

**Figure 9.** Three spatial (S1–S3) and two temporal task-redundant muscle networks (T1–T2) were empirically identified and extracted across participants and task parameters from dataset 2 using the network information framework (NIF) pipeline (panels **A, B**) (*Ó' Reilly and Delis, 2022*; *Hilt et al., 2018*). (Panel **C**) Activation coefficients are presented to the right of the networks, indicating their task parameter-specific scaling averaged across participants. Human body models accompanying each spatial network illustrate their respective submodular structure with node colour and size and edge width indicating community affiliation (*Blondel et al., 2008*), network centrality and connection strength, respectively (*Makarov et al., 2015*; *Benzi and Klymko, 2013*).

The online version of this article includes the following figure supplement(s) for figure 9:

**Figure supplement 1.** Three spatial (S1–S3) and two temporal task-redundant muscle networks (T1–T2) were empirically identified and extracted across participants and task parameters from dataset 1 using the network information framework (NIF) pipeline (panels **A, B**) (*Ó' Reilly and Delis, 2022*; *Delis et al., 2014*).

The temporal networks from datasets 1 and 2 captured mostly co-activations from movement onset – mid-movement and movement cessation, indicating that some co-contraction mechanisms were consistently task-irrelevant across trials. The temporal networks for dataset 2 were more diffuse compared to dataset 1, probably reflecting the more variable role of passive forces in generating movements to different heights captured in this dataset's experimental design (*Hilt et al., 2018*; *Hardesty et al., 2020*). Furthermore, this co-contraction mechanism was more parsimoniously represented as a single network in dataset 3 (T1 in *Figure 8B*), where passive forces in contrast likely played a consistently resistive role during locomotion. Interestingly, T1 for dataset 3 corresponded equivalently high for all three task spaces when corresponding with S2 which consisted of upper-leg extensors (see *Figure 8—figure supplement 1*). Muscle couplings indicative of agonist–antagonist pairings were also identified as separate subnetworks in S3 of dataset 3 (*Figure 8A*). More specifically, their functional segregation appeared to be based on their distinct functional roles in forward propulsion (red nodes) and deceleration (blue nodes) during the mid-stance phase of gait, as indicated by the prominent correspondence with T2 across task spaces (see *Figure 8—figure supplement 1*). This finding reflects the consistent agonistic and antagonistic contributions of muscular interactions across locomotive tasks.

The gross motor function of muscle couplings was another characteristic of task-irrelevant muscle couplings that pervaded across the datasets analysed here. For instance, AD had a central role in S2 of dataset 1 while also displaying a unique pattern of connectivity with tibial musculature in S3 of dataset 2. Similarly, GlutM had a central role in S1 of dataset 3 (*Figure 8A*). We further found a common pattern of task-irrelevant connectivity in S2 across datasets, namely the musculature about a hinge joint (elbow in datasets 1 and 2, knee in dataset 3) coupled with proximal shoulder or hip musculature, indicative of their biomechanical affordances. Finally, the passive, left arm was connected with the tibial musculature of S3 in dataset 2 (green nodes *Figure 7A*). To probe the underlying function of this connectivity in the left arm, we inspected the original EMG signals. We observed periodic,

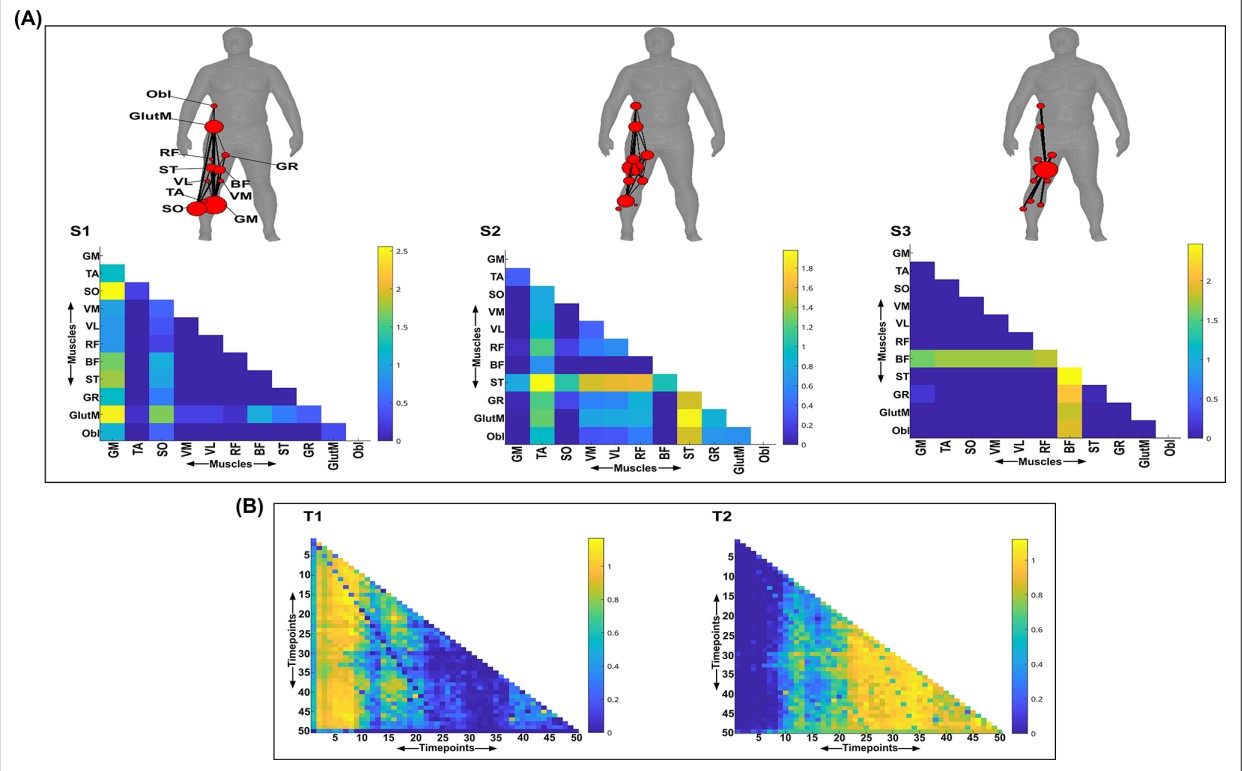

**Figure 10.** Three spatial (S1–S3) and two temporal task-redundant muscle networks (T1–T2) were empirically identified and extracted across participants and task parameters from dataset 3 using the network information framework (NIF) pipeline (**A, B**) (**Ó' Reilly and Delis, 2022**; **Camargo et al., 2021**). Activation coefficients are presented in supplementary material 1, indicating their task parameter-specific scaling averaged across participants in the dynamic, inertial motion unit (IMU), and kinematic spaces. Human body models accompanying each spatial network illustrate their respective submodular structure with node colour and size and edge width indicating community affiliation (**Blondel et al., 2008**), network centrality and connection strength, respectively (**Makarov et al., 2015**; **Benzi and Klymko, 2013**).

The online version of this article includes the following figure supplement(s) for figure 10:

**Figure supplement 1.** Task-redundant activation coefficients (dataset 3) (**Delis et al., 2014**).

tonic activations across tasks, reflective of reciprocal inhibition of contralateral limb musculature that enables unilateral movement (**Cincotta and Ziemann, 2008**).

## Task-redundant muscle couplings

To characterise the functional role of task-relevant muscle couplings, we employed a higher-order information-theoretic concept known as co-information (co-I) (**Ince et al., 2017**; **McGill, 1954**; **Schyns et al., 2020**). This metric quantifies the MI between three random variables and may take on positive values (net synergistic) and negative values (net redundant) (see **Figure 13** of '*Materials and methods*'). Co-I quantifies the task-relevant information shared between $[m_x, m_y]$ independently of the information generated by task-irrelevant muscular interactions. In doing so, it also defines the functional relationship between $[m_x, m_y]$ overall as redundant or complementary. Following the quantification of co-I for all $[m_x, m_y]$ and corresponding $\tau$ (pink area in orange-and-pink intersection **Figure 1.3B**), we parsed the negative values indicating redundancy into a separate matrix and rectified them. In **Figures 9 and 10**, we illustrate the output following the extraction of task-redundant space–time muscle networks from datasets 2 and 3 across tasks and participants, respectively, while the output for dataset 1 is presented in **Figure 9—figure supplement 1**. In the co-I formulation, task-redundant muscle couplings can be interpreted as muscle couplings that overall shared a common task-relevant functional role. For example, with reference to **Figure 9** here, muscles in the networks presented in S1 (**Figure 9A**) carry redundant information about the movement endpoint (**Figure 9C**) with the temporal profile T1 (**Figure 9B**) whereas S3 (**Figure 9A**) contains muscle networks carrying redundant information about the starting point (**Figure 9C**) with the same temporal profile T1 (**Figure 9B**).

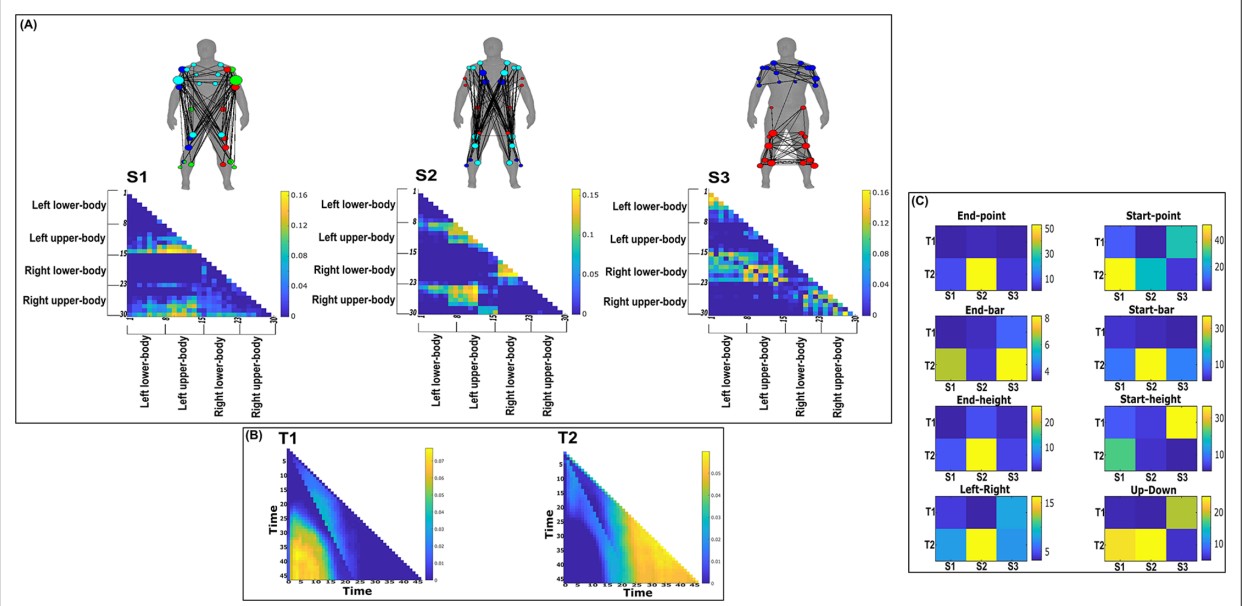

**Figure 11.** Three spatial (S1–S3) and two temporal task-synergistic muscle networks (T1–T2) were empirically identified and extracted across participants and task parameters from dataset 2 using the network information framework (NIF) pipeline (panels **A, B**) (*Ó' Reilly and Delis, 2022*; *Hilt et al., 2018*). (Panel **C**) Activation coefficients are presented to the right of the networks, indicating their task parameter-specific scaling averaged across participants. Human body models accompanying each spatial network illustrate their respective submodular structure with node colour and size and edge width indicating community affiliation (*Blondel et al., 2008*), network centrality and connection strength, respectively (*Makarov et al., 2015*; *Benzi and Klymko, 2013*).

The online version of this article includes the following figure supplement(s) for figure 11:

**Figure supplement 1.** Three spatial (S1–S3) and two temporal task-synergistic muscle networks (T1–T2) were empirically identified and extracted across participants and task parameters from dataset 1 using the NIF pipeline (panels **A, B**) (*Ó' Reilly and Delis, 2022*; *Delis et al., 2014*).

Both dataset 1 (*Figure 9—figure supplement 1*) and dataset 3 (*Figure 10*) outputs display similar patterns of muscle couplings at the same spatial scale of an individual, task-relevant limbs' musculature, with an emphasis on the coupling of specific muscles with all other muscles. For dataset 1, FE, BI, and BR displayed this integrative pattern across S1–S3, respectively, while BF, TA, and ST demonstrated this pattern in dataset 3 also. The muscle networks encapsulated several functionally interpretable couplings such as the agonist–antagonist pairings of the BI and TM of S2 (dataset 1, *Figure 9—figure supplement 1A*) and the task redundancy of ankle dorsi-flexors, and knee/hip flexors during sloped walking for example in S2 of dataset 3 (*Figure 10A*; *Pickle et al., 2016*). The functional interpretation of these muscle connectivity patterns was in line with the extracted task-specific activations. For instance, S2 of dataset 1 was modulated most prominently by reaching direction when corresponding with T2, commensurate with the biomechanical affordances of this upper-arm muscle network. Furthermore, S2 of dataset 3 was specifically modulated by the right-thigh kinematic marker along the *y*-axis (up–down direction) for both T1 and T2 (see *Figure 10—figure supplement 1*). The centrality of task-redundant muscle couplings in datasets 1 and 3 suggests particular muscle activations drive the task-specific variations in the reaching arm and stepping leg muscle activities towards a common behavioural goal. It is also worth noting that the magnitude of these functional connectivity patterns appeared to be proportional to anatomical distance, as evidenced by the magnitude of connection strengths, a finding supportive of previous related research (*Kerkman et al., 2018*).

Meanwhile at the greater spatial scale of dataset 2 (*Figure 9A*), task-redundant muscle couplings were anatomically compartmentalised to the upper- and lower-body. This functional segregation was emphasised at the subnetwork level also, where the upper- and lower-body musculature of S3 for instance formed distinct submodules (blue and red nodes). Amongst the task-specific activations in dataset 2, S1 carried redundant task information about endpoint target, -height, and up–down direction when corresponding with T1. T2 for dataset 2 on the other hand contained mostly temporally proximal dependencies along the diagonal, suggestive of co-contraction mechanisms, which

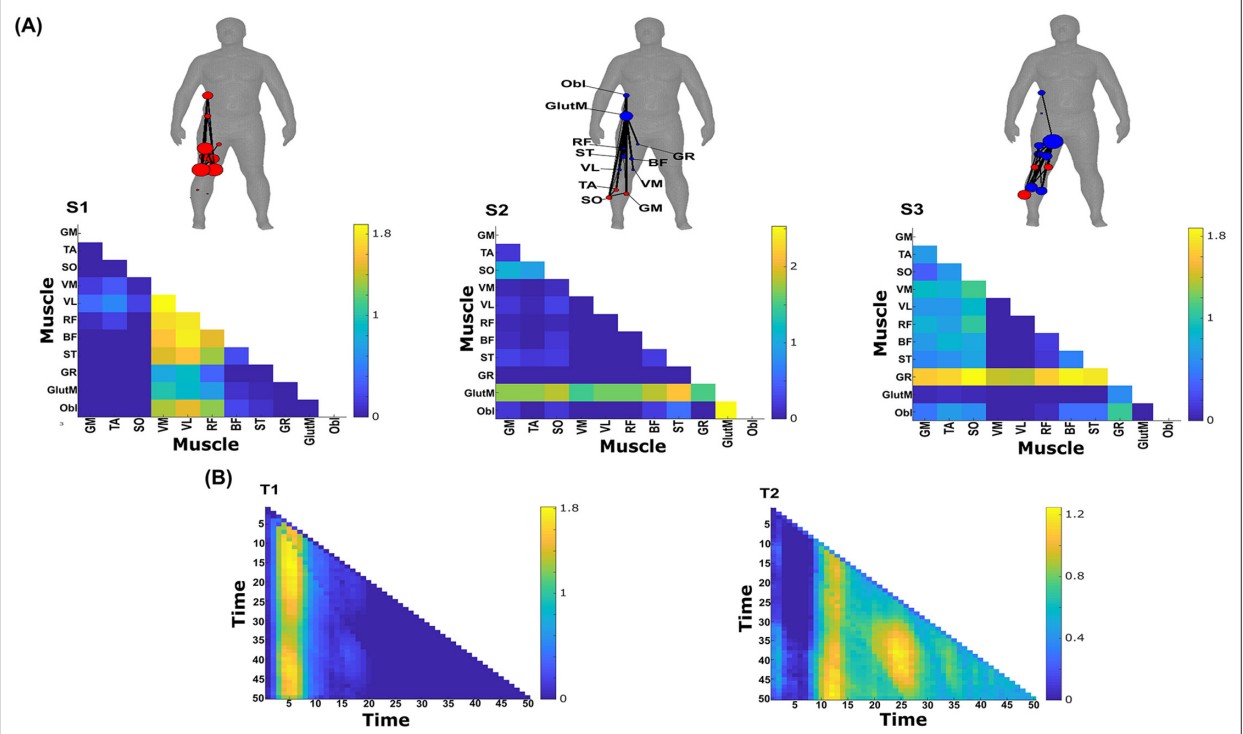

**Figure 12.** Three spatial (S1–S3) and two temporal task-synergistic muscle networks (T1–T2) were empirically identified and extracted across participants and task parameters from dataset 3 using the network information framework (NIF) pipeline (panels **A, B**) (*Ó' Reilly and Delis, 2022*; *Camargo et al., 2021*). Activation coefficients are presented in supplementary material 1, indicating their task parameter-specific scaling averaged across participants in the dynamic, inertial motion unit (IMU), and kinematic spaces. Human body models accompanying each spatial network illustrate their respective submodular structure with node colour and size and edge width indicating community affiliation (*Blondel et al., 2008*), network centrality and connection strength, respectively (*Makarov et al., 2015*; *Benzi and Klymko, 2013*).

The online version of this article includes the following figure supplement(s) for figure 12:

**Figure supplement 1.** Task-synergistic activation coefficients (dataset 3) (*Delis et al., 2014*).

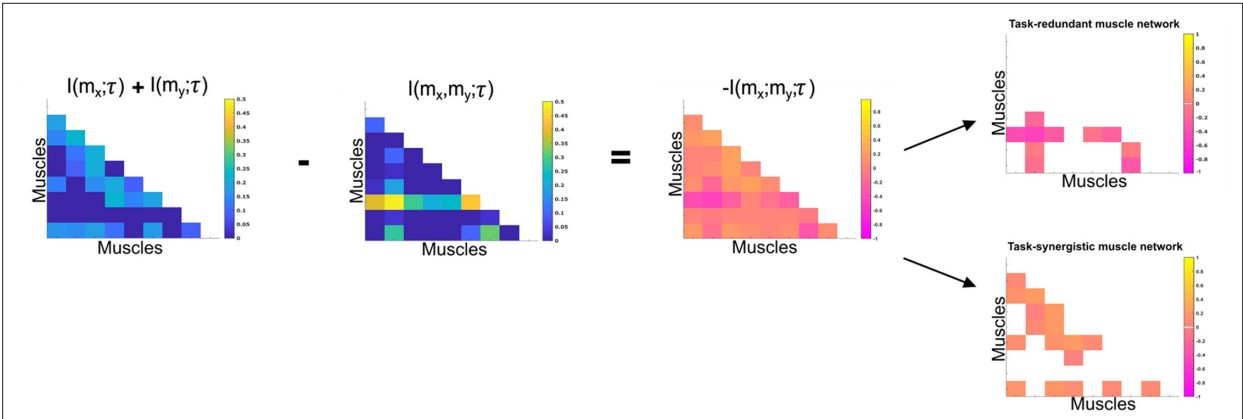

**Figure 13.** Co-information (Co-I) determines the difference between the sum total information shared with $\tau$ in $m_x$ and $m_y$ when observed separately and the information shared with $\tau$ when they are observed together. The adjacency matrices show how this calculation is carried out for all unique $[m_x, m_y]$ combinations. Redundant and synergistic muscle couplings are then separated into two equivalently sized networks. The accompanying colour bars indicate the values present in the adjacency matrix.

became more diffuse near movement cessation. These endpoint trajectory and co-contraction related temporal patterns were qualitatively similar to T1 and T2 of both datasets 1 and 3, respectively (see *Figure 10* and *Figure 9—figure supplement 1*, respectively).

## Task-synergistic muscle couplings

Similarly, we isolated task-synergistic muscle couplings by parsing, instead, the positive co-I values from the computations conducted across all $[m_x, m_y]$ and corresponding $\tau$ into a sparse matrix where all redundant couplings were set to zero (see *Figure 13* of '*Materials and methods*'). Task-synergistic muscle couplings here can be interpreted as a $[m_x, m_y]$ pair that provide complementary (i.e. functionally dissimilar) task information, thus more information is gained by observing $[m_x, m_y]$ together rather than separately (orange area in orange-and-pink intersection *Figure 1.3B*). In *Figures 11 and 12*, we illustrate the task-synergistic space–time muscle networks from datasets 2 and 3, respectively (dataset 1 output is presented in *Figure 11—figure supplement 1*).

Across datasets, muscle networks could be characterised by the transmission of complementary task information between functionally specialised muscle groups, many of which were identified among the task-redundant representations (*Figures 9 and 10* and *Figure 9—figure supplement 1*). The most obvious example of this is the S3 synergist muscle network of dataset 2 (*Figure 11*), which captures the complementary interaction between task-redundant submodules identified previously (S3, *Figure 9*). A particularly consistent structural feature was the emphasis of an individual muscles' connectivity with all other muscles which was evident among synergistic couplings in dataset 3 (see VM, VL, and RF of S1, GR, TA, and SO of S2, and GlutM of S3 in *Figure 12A*). This structural similarity demonstrates that parallel and synchronous exchanges of redundant and synergistic task information underlie task-specific variations across trials (e.g. S3 of dataset 2 in *Figure 9A* and *Figure 11A*).

**Table 1.** A summary table illustrating the findings from an examination of the generalisability of the muscle networks extracted from each dataset.

The spatial and temporal representations extracted from the full input data in each muscle-task information subspace were compared using Pearson's correlation against functionally similar representations extracted from a subset of the input data.

| | Dataset 1 | | | |
| --- | --- | --- | --- | --- |
| | Spatial | | Temporal | |
| | *Participants* | *Tasks* | *Participants* | *Tasks* |
| Task-redundant | 0.95 ± 0.23 | 0.89 ± 0.28 | 0.98 ± 0.19 | 0.89 ± 0.27 |
| Task-synergistic | 0.95 ± 0.23 | 0.73 ± 0.26 | 0.98 ± 0.19 | 0.91 ± 0.3 |
| Task-irrelevant | 0.84 ± 0.1 | 0.85 ± 0.1 | 0.79 ± 0.15 | 0.95 ± 0.04 |

| | Dataset 2 | | | |
| --- | --- | --- | --- | --- |
| | Spatial | | Temporal | |
| | *Participants* | *Tasks* | *Participants* | *Tasks* |
| Task-redundant | 0.82 ± 0.42 | 0.94 ± 0.27 | 0.99 ± 0.03 | 0.96 ± 0.28 |
| Task-synergistic | 0.79 ± 0.37 | 0.96 ± 0.25 | 0.92 ± 0.3 | 0.99 ± 0.1 |
| Task-irrelevant | 0.84 ± 0.15 | 0.99 ± 0.01 | 0.93 ± 0.1 | 0.99 ± 0.06 |

| | Dataset 3 | | | | | |
| --- | --- | --- | --- | --- | --- | --- |
| | Spatial | | | Temporal | | |
| | *Kinematics* | *Dynamics* | *IMU* | *Kinematics* | *Dynamics* | *IMU* |
| Task-redundant | 0.92 ± 0.28 | 0.91 ± 0.32 | 0.86 ± 0.31 | 0.93 ± 0.21 | 0.92 ± 0.24 | 0.87 ± 0.29 |
| Task-synergistic | 0.97 ± 0.16 | 0.9 ± 0.27 | 0.96 ± 0.19 | 0.99 ± 0.04 | 0.98 ± 0.13 | 0.95 ± 0.18 |
| Task-irrelevant | 0.9 ± 0.34 | 0.83 ± 0.37 | 0.8 ± 0.33 | 0.99 ± 0.04 | 0.98 ± 0.1 | 0.97 ± 0.15 |

Interestingly, despite the similarity between the redundant and synergistic muscle networks, the way they are combined to encode task information differs depending on the type of interaction (synergistic vs. redundant, e.g. panel C of *Figure 11* in comparison to panel C of *Figure 9*).

Concerning the temporal activations of these networks, the task-synergistic structure of dataset 2 (*Figure 11B*) was also relatively unchanged compared to the task-redundant structure (*Figure 9B*). This suggests that the task end-goal- and co-contraction-related mechanisms provided both redundant and synergistic task information concurrently during whole-body reaching movements. In contrast, a different view of synergistic information exchange is provided in datasets 1 and 3 (*Figure 11—figure supplement 1* and *Figure 12*, respectively), where T1 and T2 consist of more idiosyncratic activations that together appear to reflect the task end-goal-related patterns found elsewhere. More specifically, in both datasets we found two distinct patterns of task end-goal-related activity where early and late timepoints during movement initiation operated in parallel to provide complementary task information (see *Figure 12B* and *Figure 11—figure supplement 1*).

## Generalisability of the extracted space–time muscle networks

To ascertain the generalisability of the extracted representations presented here beyond any subset of the input data, we conducted a similarity analysis through a leave-*n*-out cross validation procedure. In more detail, we compared the space–time networks extracted from the full dataset (illustrated in *Figures 4–9* and corresponding supplementary materials) to the networks extracted from a subset of the data (see '*Materials and methods*'). Across datasets, a high level of concordance was found on average (~0.9 correlation, see *Table 1*). This trend was evident across all datasets and for task-redundant, -synergistic, and -irrelevant spatial and temporal networks. Dataset 1 and 2 findings demonstrate that the extracted networks are generalisable beyond any individual participant or task. Dataset 3 results go further by demonstrating that the extracted patterns are generalizable beyond any randomly selected and randomly sized subset of the input data. The highest correlations on average were consistently among temporal networks, replicating previous findings (*Ó' Reilly and Delis, 2022*; *Delis et al., 2018b*; *Brambilla et al., 2022*). Although the spatial networks demonstrated a lower average correlation, this was substantially higher compared to previous applications of datasets 1 and 2 (*Ó' Reilly and Delis, 2022*; *Delis et al., 2014*; *Delis et al., 2018b*), suggesting that the inclusion of task parameters here captures the inter-participant differences more effectively. When comparing representations extracted from each continuous task space in dataset 3, kinematic features consistently had the highest average correlation and lowest variability compared to dynamic and IMU feature spaces. These findings support our change in the interpretation of the extracted activation coefficients away from conventional approaches where representational biases towards particular participants and/or tasks are inferred.

To further probe how the underlying assumption of an all-to-all correspondence between spatial and temporal representations made by sNM3F influenced the generalisability of the extracted networks, we compared its performance to non-negative Canonical-Polyadic (CP) tensor decomposition, which assumes an opposing one-to-one correspondence between components. We found that although CP has demonstrated a considerable capacity to de-mix neural data into simplified and interpretable low-dimensional components (*Williams et al., 2018*), its application here resulted in poor generalisability of the extracted patterns (~0.5 correlation on average). This finding suggests that the all-to-all correspondence implemented by sNM3F identifies a more generalizable representation and should be favourably considered in future applications of this framework.

## Discussion

The aim of the current study was to dissect the muscle synergy concept and offer a novel, more nuanced perspective on how muscles '*work together*' to achieve a common behavioural goal. To do so, we introduced a computational approach based on the NIF pipeline (*Figure 3*), enabling the effective decomposition of muscle interactions and the comprehensive description of their functional roles. Through the direct inclusion of task parameters in the extraction of muscle synergies, a novel perspective was produced where muscles '*work together*' not just towards a common task-goal, but also concomitantly towards complementary task objectives and lower-level functions irrelevant for overt task performance. The functional architectures we uncovered were comprised of distributed

subnetworks of synchronous muscle couplings and driven by distinct temporal patterns operating in parallel. Example applications to simple real and simulated EMG datasets revealed the additional capabilities provided by the proposed framework to current muscle synergy analysis in terms of functional and physiological relevance and interpretability. When applied to large-scale data, the proposed methodology extracted representations scale-invariant to dataset complexity and motor behaviours whilst being generalizable beyond any subset of the data. We thus present this framework as a useful analytical approach for mechanistic investigations on large-scale neural data through this novel perspective to the muscle synergy.

The '*muscle synergy*' is a major guiding concept in the motor control field (***Bruton and O'Dwyer, 2018***; ***Bizzi and Cheung, 2013***; ***Berret et al., 2019***; ***Bernstein, 1966***; ***Latash, 2008***; ***Cheung and Seki, 2021***), providing a conceptual framework to the problem of motor redundancy that centres around the general notion of '*working together*'. In its current conceptualisation, '*working together*' describes how the nervous system functionally groups muscles in a task-dependent manner through common neural drives to simplify movement control. This idea has undergone continued refinement since its early conception (***Bruton and O'Dwyer, 2018***; ***Bernstein, 1966***; ***Latash, 2008***), with a notable progression being the introduction of the qualifying attributes: a sharing pattern, reciprocity, and task dependence (***Latash, 2008***). Nonetheless, recent influential works revealing other important mechanisms for the simplification of motor control highlight the generality of the current perspective offered by this concept (***Ronzano et al., 2021***; ***Hug et al., 2021***; ***Hug et al., 2023***; ***Alessandro et al., 2020***; ***Valero-Cuevas et al., 2009***; ***Nazarpour et al., 2012***; ***Todorov and Jordan, 2002***; ***d'Avella and Bizzi, 2005***; ***Cheung et al., 2005***). We thus sought to provide greater nuance to the notion of '*working together*' by defining motor redundancy and synergy in information-theoretic terms (***Bernstein, 1966***; ***Schneidman et al., 2003***). In our framework, redundancy and synergy are terms describing functionally similar and complementary motor signals, respectively, introducing a new perspective that is conceptually distinct from the traditional view of muscle synergies as a solution to the motor redundancy problem (***Bruton and O'Dwyer, 2018***; ***Bernstein, 1966***; ***Latash, 2008***). In this new definition of muscle interactions in the task space, a group of muscles can '*work togethe*r' either synergistically or redundantly towards the same task. In doing so, the perspective instantiated by our approach provides novel coverage to the partitioning of task-relevant and -irrelevant variability implemented by the motor system along with an improved specificity regarding the functional roles of muscle couplings (***Valero-Cuevas et al., 2009***; ***Nazarpour et al., 2012***; ***Todorov and Jordan, 2002***). Our framework emphasises not only the role of functionally redundant muscle couplings that result from the underlying degeneracy of the motor system, but also of complementary, synergistic dependencies that are important for communication and integration across specialised neural circuitry (***Nigam et al., 2019***; ***Luppi et al., 2022***). Thus, the present study aligns the muscle synergy concept with the current mechanistic understanding of the nervous system whilst offering an analytical approach amenable to the continued advances in large-scale data capture (***Krakauer et al., 2017***; ***Urai et al., 2022***).

Among current approaches to muscle synergy analysis, the established synchronous, temporal, and time-varying muscle synergy models are understood to each characterise unique motor features (***Chiovetto et al., 2013***). More specifically, the synchronous model captures agonist–antagonist muscle pairings, the temporal model decomposes EMG signals into functionally distinct temporal phases and the task-specific modulation of spatiotemporal invariants are quantified in the time-varying model. In a unifying framework, here we quantify space–time muscle networks that concurrently capture many of these salient motor features in a holistic and principled way whilst mapping their functional consequences to motor behaviour. These salient features included, among others, agonist–antagonist pairings and functionally meaningful inter-limb couplings that consistently appeared across task-redundant, -synergistic, and -irrelevant spaces. Thus, in dissecting the muscle interactions governing coordinated movement, our framework revealed their parallel and synchronous processing of functionally similar, complementary, and task-irrelevant information. This insight aligns with several recent works also demonstrating this distributed neural architecture of parallel information processing units (***Macpherson et al., 2021***; ***Nigam et al., 2019***; ***Luppi et al., 2022***).

Our framework also revealed novel characteristics of the motor system. For instance, the task-redundant and -synergistic networks we extracted appeared to be structured around the coupling between a prime-mover muscle and several supporting muscles, supporting recent work showing the nonhomogeneous sharing of neural drives within modules (***Del Vecchio et al., 2022***). These novel

spatial characteristics were driven by parallel temporal patterns representing endpoint trajectory and co-contraction-related mechanisms, an insight supportive of recent work showing their parallel innervation (*Ronzano et al., 2021*; *Hardesty et al., 2020*; *Borzelli et al., 2022*). Together, these representations encapsulated the functional interplay between task end-goal requirements and biomechanical affordances, a dynamic frequently highlighted in object manipulation experiments (*Sartori et al., 2011*). In other words, the task-relevant networks reflected how muscles 'work together' both redundantly and synergistically towards a desired end-goal state whilst, in parallel, continually controlling the present trajectory of the system. Meanwhile, the task-irrelevant networks demonstrated that these muscle couplings also work concomitantly towards lower-level objectives assigned mostly during the transition to this desired end state. Although distinguishing task-irrelevant muscle couplings may capture artifacts such as EMG crosstalk, our results convey several physiological objectives of muscles including gross motor functions (*Dounskaia et al., 2020*), the maintenance of internal joint mechanics and reciprocal inhibition of contralateral limbs (*Alessandro et al., 2020*; *Cincotta and Ziemann, 2008*). Thus, task-irrelevant muscle interactions reflect both biomechanical- and task-level constraints that provide a structural foundation for task-specific couplings. The separate quantification of these muscle interaction types opens up novel opportunities in the practical application of muscle synergy analysis, as demonstrated in the current study through the identification of a significant predictor of motor impairment post-stroke from single trials (*Berret et al., 2019*; *Alessandro et al., 2013*; *Santello et al., 2016*). For instance, these distinct representations may encapsulate different neural substrates that can be specifically assessed at the muscle-level for the purpose of bodily restoration and augmentation (*Dominijanni et al., 2021*). Uncovering their neural underpinnings is an interesting topic for future research.

Indeed, in future work, we aim to complement this study's combinatorial perspective to the muscle synergy by dissecting the unique contribution of individual muscles to motor behaviour and how they may work independently towards task performance (see magenta and cyan intersections in *Figure 1.3B*). More broadly, our work here parallels related information-theoretic approaches to decomposing task-relevant brain activity (*Schyns et al., 2020*; *Delis et al., 2018a*), whilst addressing a current research gap across the neurosciences in effective methods to mapping large-scale, spatio-temporal neural activity to behaviour (*Krakauer et al., 2017*; *Urai et al., 2022*). Future applications of this framework should include large-scale, multi-modal data captured from participants performing a wide range of natural behaviours.

In sum, this study introduced a novel perspective to the muscle synergy concept and a computational framework to extract muscle couplings that map their pairwise contributions to motor behaviour. We suggest that this approach offers novel research opportunities for investigating the underlying neural constraints on motor behaviour and the fundamental structure–function relationships generated by agent–environment interactions (*Cheung and Seki, 2021*; *Adams et al., 2013*).

## Materials and methods
### Quantifying muscle couplings in the task space

To quantify muscle couplings we used MI, a non-linear measure of dependence that captures any type of relationship between random variables. Here to estimate MI, we used a Gaussian copula-based approximation (*Ince et al., 2017*). This semi-parametric estimator exploits the equivalence between MI and the negative entropy of the empirical copula ($c$), a function that maps a multivariate set (e.g. $[m_x, m_y]$ representing activities of muscles X and Y) to their joint distribution (*Equation 1.1*; *Ince et al., 2017*).

$$I\left(m_x; m_y\right) = -H\left(c\right) \tag{1.1}$$

Thus, to determine task-irrelevant muscle couplings ($I\left(m_x, m_y|\tau\right)$), we conditioned the negative entropy of the empirical copula for $[m_x, m_y]$ with respect to a task variable $\tau$ (*Equation 1.2*). As mentioned, $[m_x, m_y]$ are continuous vectors composed of individual muscle amplitudes at specific timepoints across trials while $\tau$ is a corresponding discrete (e.g. movement direction for datasets 1 and 2) or continuous (e.g. movement kinematics in dataset 3) task parameter. For discrete task variables, $\tau$ takes one value for each trial and the MI is calculated across trials using a Gaussian mixture model (*Ince et al., 2017*). In the case of continuous task variables, $\tau$ varies in time within a specific

trial. Thus, we compute MI at each timepoint $t$ using the muscle activity $m_x(t)$ and the task variable value $\tau(t)$ at this time point using a closed-form parametric estimator (**Ince et al., 2017**).

$$I\left(m_x; m_y | \tau\right) = -H\left(c | \tau\right) \tag{1.2}$$

To evaluate the task-relevance of the identified muscle couplings, we used a higher-order information-theoretic measure known as co-I (**Equation 1.3**; **Figure 11**), which quantifies the relationship between three random variables, here $[m_x, m_y]$, and $\tau$. Co-I implements the inclusion–exclusion principle of combinatorics (**McGill, 1954**), whereby the sum of MIs between individual $m$ vectors and $\tau$ $(I\left(m_x; \tau\right) + I\left(m_y; \tau\right))$ is compared against their composite MI $(I\left(m_x, m_y; \tau\right))$ as follows:

$$-I\left(m_x; m_y; \tau\right) = I\left(m_x; \tau\right) + I\left(m_y; \tau\right) - I\left(\left[m_x, m_y\right]; \tau\right) \tag{1.3}$$

Negative $I\left(m_x; m_y; \tau\right)$ corresponds to a net redundant coupling between $[m_x, m_y]$ about $\tau$ while positive $I\left(m_x; m_y; \tau\right)$ indicates a net synergy. To analyse these distinct couplings separately, we parsed redundant and synergistic $I\left(m_x; m_y; \tau\right)$ into two equivalently sized matrices and rectified the redundant couplings to make them suitable for non-negative dimensionality reduction.

Then, to produce a multiplexed view of the muscular interactions across trials, we iterated these MI computations over all unique combinations of $[m_x, m_y]$ and $\tau$. The resulting MI estimates collectively form $A$, a symmetric adjacency matrix (i.e. $A^T A = I$) that represents the functional connectivities between all muscle activations (**Figure 10**). When repeated across all available task variables $\tau$ and participants, $A$ is of dimension $[\textit{No. of muscle pairs} \times \textit{No. of timepoint pairs} \times [\textit{No. of } \tau \times \textit{No. of participants}]]$. Thus, by applying network-theoretic statistical tools to $A$, we can identify functional modules carrying the same type of (redundant/synergistic) task information (**Figure 2B**).

## Estimating statistical significance of muscle couplings

To isolate statistically significant dependencies, we applied a modified percolation analysis to each $A$ (**Gallos et al., 2012**). This method sparsifies functional connectivities in $A$ with respect to its percolation threshold ($P_c$). $P_c$ is a critical value that specifies the probability of a nodes' occupation in $A$ with respect to the networks size. In random networks, a '*giant component*' comprised of long-range connections exists above $P_c$ but disappears as $P_c$ tends to zero (**Bunde and Havlin, 2012**), while it is thought that living systems optimise adaptability by fluctuating around $P_c$ in a state of self-organised criticality (**Bak et al., 1987**). Preliminary testing of this method showed it to be at least equivalent to permutation-testing each MI value in the network and thus, much more computationally efficient. $P_c$ was therefore iteratively specified for each layer of $A$ relative to equivalently sized random networks and utilised to remove insignificant network edges up to a stopping-point where this giant component begins to become affected (**Figure 3C**). This procedure was carried out for each layer of $A$ separately configured as muscle-wise couplings across temporal scales (i.e. a 3D tensor of dimension $[\textit{No. of muscle}] \times [\textit{No. of muscle}] \times [\textit{No. of timepoint pairings} \times \textit{No. of } \tau \times \textit{No. of participants}])$ and vice versa as timepoint-wise couplings across spatial scales (**Figure 3D**). The separate sparsification of each individual network layer in both alternative network configurations produced discrepancies in the output, as some connections were found to be significant in only one domain. To ameliorate this discrepancy, we employed a conservative heuristic where dependencies must be significant in both space and time to be included in the final input matrix for dimensionality reduction. Thus, the sparsified input matrices to dimensionality reduction were comprised of significant spatiotemporal task-redundant, -synergistic, or -irrelevant muscle couplings.

## Model-rank specification

To determine the optimal number of modules to extract, we implemented two alternative community detection algorithms generalised to multiplex networks (**Blondel et al., 2008**; **Mucha et al., 2010**; **Didier et al., 2018**). Both forms seek to optimise a modularity criterion known as the $Q$-statistic that quantifies the proportion of within-community network edges compared to what would be expected from a network consisting of random connections (**Newman and Girvan, 2004**). More specifically, for a particular division of a single layer network (**Equation 2.1**), let $\delta\left(g_i, g_j\right)=1$ if nodes $i$ and $j$ belong to the same group ($g$) and 0 otherwise and $A_{ij}$ be the number of edges between nodes $i$ and $j$. The equivalent of $A_{ij}$ from a randomised network ($P_{ij}$) is expected to be $\frac{k_i k_j}{2m}$ (Newman–Girvan null

model) (**Newman and Girvan, 2004**), where $k_i$ and $k_j$ are the node degrees and $m = \frac{1}{2}\sum_{ij} A_{ij}$ . The typical output of the $Q$-statistic is found within the range [0,1] with 1 indicating maximum modularity (**Newman and Girvan, 2004**).

$$Q_{monolayer} = \frac{1}{4m}\sum_{ij}\left(A_{ij} - P_{ij}\right)\delta\left(g_i, g_j\right) \tag{2.1}$$

In its generalised multilayer form, the $Q$-statistic is given an additional term to consider couplings between layers $l$ and $r$ with intra- and inter-layer resolution parameters $\gamma$ and $\omega$ (**Equation 2.2**). Here, $\mu$ is the total edge weight across the network and $\gamma$ and $\omega$ were set to 1 in the current study for classical modularity (**Mucha et al., 2010**), thus removing the need for any hyperparameter tuning.

$$Q_{multilayer} = \frac{1}{2\mu}\sum_{ijlr}\left[\left(A_{ijl} - \gamma_l P_{ijl}\right)\delta_{lr} + \delta_{ij}\omega_{jlr}\right]\delta\left(g_{il}, g_{jr}\right) \tag{2.2}$$

We chose to implement two complementary model-rank specification approaches to address limitations related to stochasticity and scalability present in the multilayer formulation and the inability to consider inter-layer dependencies present in the mono-layer formulation (**Hug et al., 2023**; **Magnani et al., 2022**). To apply these algorithms to our data, we grouped the set of $A$ into multiplex networks configured with respect to spatial or temporal scales (**Figure 3D**). We then applied these algorithms to both space–time network configurations for individual participant/tasks. This procedure generated a binary adjacency matrix from the resulting community partition vector in each case where 1 indicated the nodes belonged to the same community and 0 otherwise (**Didier et al., 2018**). Following a consensus-based approach (**Lancichinetti and Fortunato, 2012**), we then grouped these binary adjacency matrices into a new multiplex network and re-applied the two alternative community detection algorithms to find an optimal spatial and temporal model-rank (**Lancichinetti and Fortunato, 2012**).

### Extraction of low-dimensional representations

Following the specification of an optimal model-rank in the spatial and temporal domains, we used these values as input parameters into dimensionality reduction (**Figure 3E**). To extract a low-dimensional representation of motor behaviour across muscle couplings, we applied a sample-based non-negative matrix tri-factorisation method (sNM3F) with additional orthogonality constraints to the matrices consisting of vectorised and concatenated $A$ (**Delis et al., 2014**). More specifically, we decomposed the input three-mode tensor $A$ of dimension [K = No. of unique muscle pairs ($m$) × (L = No. of time-sample pairs ($t$) × No. of participants + tasks)] into a set of spatial ($V$) and temporal ($W$) factors and the participant- and task-specific weighting coefficients ($S$) reflecting the amount of information carried by each combination of spatial–temporal factors for each participant and task parameter (**Equation 3.1**). In **Equation 3.2**, this factorisation is also illustrated in vector sum notation for a single participant and task variable:

$$A \approx WSV \tag{3.1}$$

$$\begin{pmatrix} \left(m^1 t^1\right) & \cdots & \left(m^1 t^L\right) \\ \vdots & \ddots & \vdots \\ \left(m^K t^1\right) & \cdots & \left(m^K t^L\right) \end{pmatrix} \approx \begin{pmatrix} w_i^{t^1} & \cdots & w_i^{t^L} \end{pmatrix} \cdot \begin{pmatrix} s_{v_1}^{w^1} \dots s_{v_j}^{w^1} \\ s_{v_1}^{w^i} \dots s_{v_j}^{w^i} \end{pmatrix} \cdot \begin{pmatrix} v_j^{m^1} \\ \vdots \\ v_j^{m^K} \end{pmatrix} + residuals \tag{3.2}$$

### Examining the generalisability of extracted representations

To determine the generalisability of the extracted space–time muscle networks, we implemented a representational similarity analysis where we compared representations extracted from the full datasets 1–3 to equivalent networks extracted from a subset of the respective datasets. We computed the similarity between pairs of representations using Pearson's correlation.

For datasets 1 and 2 (see below), we removed from the input data an individual participant or task at a time and compared the similarity of the decomposition outputs with those obtained from the full

dataset. We repeated this for all participants and task variables and reported the average similarity as a measure of robustness of the decomposition.

For dataset 3 (see below), due to the greater number of participants and task parameters, we implemented a more stringent examination. More specifically, we firstly extracted representations from each task space individually and, using these representations as a reference, compared them against functionally similar outputs after removing randomly sized portions of randomly selected vectors in the input data (up to the no. of column vectors − 1). We repeated this procedure for 50 iterations, and computed summary statistics by converting the coefficients to Fisher's $Z$ values, computing the average and standard deviation, and then reverting these values back to correlation coefficients.

## Subnetwork analysis

To illustrate the relative importance of individual muscles within each network, we determined the total communicability ($C\left(i\right)$) of individual nodes ($i$) in each network ($A$). $C\left(i\right)$ is defined as the row-wise sum of all matrix exponentials ($e$) in the adjacency matrix ($A$) that consider the number of walks between each pair of nodes $i$ and $j$ (*Equation 4.1*; *Benzi and Klymko, 2013*; *Estrada and Hatano, 2008*):

$$C\left(i\right) = \sum_{j=1}^{N}\left[e^{A}\right]_{ij} \tag{4.1}$$

To emphasise salient functional connectivities in the spatial networks, we sparsified all dependencies with a below average network communicability and illustrated the output on the accompanying human body models (*Makarov et al., 2015*; *Estrada and Hatano, 2008*). To uncover salient subnetwork structures consisting of more closely functionally related muscle activations, we applied the monolayer community detection algorithm in *Equation 2.1* to the extracted spatial networks (*Blondel et al., 2008*; *Rubinov and Sporns, 2010*).

## Data acquisition and processing

To illustrate our framework, we applied it to three datasets of EMG signals recorded during different motor tasks. In dataset 1 (*Figure 4A*), 7 adult participants (age: 27 ± 2 years, height: 1.77 ± 0.03 m) performed table-top point-to-point reaching movements in both forward and backwards directions and at fast and slow speeds while the activity of nine muscles on the preferred right, reaching arm (finger extensors (FE), brachioradialis (BR), biceps brachii (BI), medial-triceps (TM), lateral-triceps (TL), anterior deltoid (AD), posterior deltoid (PD), pectoralis major (PE), latissimus dorsi (LD)) were captured for a total of 640 trials per participant (*Delis et al., 2014*). To enable the quantification of shared information across muscles with respect to specific task attributes, we formulated four discrete task parameters of length equal to the number of trials executed. These discrete variables represented the trial-to-trial variation in reaching direction (fwd vs. bwd), speed (high vs. low), and reaching target [P1–P8] (*Figure 3A*).

In dataset 2 (*Figure 4B*), 3 participants performed whole-body point-to-point reaching movements in various directions and to varying heights while EMG from 30 muscles (tibialis anterior, soleus, peroneus, gastrocnemius, vastus lateralis, rectus femoris, biceps femoris, gluteus maximus, erector spinae, pectoralis major, trapezius, anterior deltoid, posterior deltoid, biceps and triceps brachii) across both hemi-bodies were captured (*Hilt et al., 2018*). Like dataset 1, we formulated task parameters each representing a specific task attribute across trials (~2160 trials per participant). In this case, we formed eight discrete task parameters representing start- and endpoint target, -bar, and -height and both up–down (vertical) and left–right (horizontal) reaching directions.

Dataset 3 consisted of multiple trials from 17 participants performing level-ground walking, stair- and ramp-ascents/descents with various sub-conditions (walking speed, clockwise/counter-clockwise direction, different stair/ramp inclines, etc.) (*Figure 4C*; *Camargo et al., 2021*). These locomotion modes were performed while 11 EMG signals were captured from the right lower limb (gluteus medius (GlutM), right external oblique (Obl), semitendinosus (ST), gracilis (GR), biceps femoris (BF), rectus femoris (RF), vastus lateralis (VL), vastus medialis (VM), soleus (SO), tibialis anterior (TA), gastrocnemius medialis (GM)), *XYZ* coordinates were captured bilaterally from 32 kinematic markers and 4 IMUs and a force-plate captured accelerations and dynamic features among the lower limbs also. More detailed

breakdowns of the experimental design for each dataset can be found at their parent publications (*Delis et al., 2014*; *Hilt et al., 2018*; *Camargo et al., 2021*).

For all datasets, we processed the EMG signals offline using a standardised approach (*d'Avella and Lacquaniti, 2013*): the EMGs for each sample were digitally full-wave rectified, low-pass filtered (Butterworth filter; 20 Hz cut-off; zero-phase distortion), normalised to 1000 time-samples and then the signals were integrated over 20 time-step intervals yielding a waveform of ~50 time-steps. To match the time-series lengths, we resampled the kinematic, dynamic and IMU recordings of dataset 3 using cubic-spline interpolation to match the EMG signals.

## Additional information

### Funding

| Funder | Grant reference number | Author |
|---|---|---|
| Royal Society | RGS\R2\92224 | Ioannis Delis |
| Biotechnology and Biological Sciences Research Council | FTMA Leeds University | Ioannis Delis |

The funders had no role in study design, data collection, and interpretation, or the decision to submit the work for publication.

### Author contributions
David O'Reilly, Conceptualization, Resources, Data curation, Software, Formal analysis, Validation, Investigation, Visualization, Methodology, Writing - original draft, Writing – review and editing; Ioannis Delis, Conceptualization, Resources, Formal analysis, Supervision, Funding acquisition, Methodology, Project administration, Writing – review and editing

### Author ORCIDs
David O'Reilly http://orcid.org/0000-0002-8471-9447
Ioannis Delis https://orcid.org/0000-0001-8940-5036

Reviewer #1 (Public Review): https://doi.org/10.7554/eLife.87651.4.sa1
Reviewer #2 (Public Review): https://doi.org/10.7554/eLife.87651.4.sa2
Reviewer #3 (Public Review): https://doi.org/10.7554/eLife.87651.4.sa3
Author Response https://doi.org/10.7554/eLife.87651.4.sa4

## Additional files

### Supplementary files
• MDAR checklist

### Data availability
Dataset1 (https://figshare.com/articles/dataset/Dataset_1/25109144) and Dataset2 (https://figshare.com/articles/dataset/Dataset_2/25109156) are available at the provided repositories. Data from datasets 3 and 4 are freely available online as published datasets (*Camargo et al., 2021*; *Averta et al., 2021*).

The following datasets were generated:

| Author(s) | Year | Dataset title | Dataset URL | Database and Identifier |
|---|---|---|---|---|
| Delis I, O'Reilly D | 2024 | Dissecting muscle synergies in the task space | https://figshare.com/articles/dataset/Dataset_1/25109144 | figshare, Dataset_1/25109144 |

*Continued on next page*

*Continued*

| Author(s) | Year | Dataset title | Dataset URL | Database and Identifier |
|---|---|---|---|---|
| Delis I, O'Reilly D | 2024 | Dissecting muscle synergies in the task space | https://figshare.com/articles/dataset/Dataset_2/25109156 | figshare, Dataset_2/25109156 |

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
