## [Editor Report · eLife assessment]

The work by O'Reilly and Delis is **important** to extend the synergy ideas using methods from signal processing and information theory to cluster muscles and task parameters, thereby advancing our understanding of the modular architecture of motor control. The method is innovative, and the findings are **compelling** from theoretical and practical perspectives. The work will be of broad interest to motor control and neural engineering researchers.

---

## [Referee Report · Reviewer #1 (Public Review)]

The proposed study provides an innovative framework for the identification of muscle synergies taking into account their task relevance. State-of-the-art techniques for extracting muscle interactions use unsupervised machine-learning algorithms applied to the envelopes of the electromyographic signals without taking into account the information related to the task being performed. In this work, the authors suggest to include the task parameters in extracting muscle synergies using a network information framework previously proposed. This allows the identification of muscle interactions that are relevant, irrelevant, or redundant to the parameters of the task executed.

The proposed framework is a powerful tool to understand and identify muscle interactions for specific task parameters and it may be used to improve man-machine interfaces for the control of prostheses and robotic exoskeletons.

With respect to the network information framework recently published, this work added an important part to estimate the relevance of specific muscle interactions to the parameters of the task executed.

It is not clear how the well-known phenomenon of cross-talk during the recording of electromyographic muscle activity may affect the performance of the proposed technique and how it may bias the overall outcomes of the framework.

---

## [Referee Report · Reviewer #2 (Public Review)]

This paper is an attempt to extend or augment muscle synergy and motor primitive analyses and ideas with addition of task-driven measures. The authors' idea is to use information metrics (mutual information, co-information) in 'synergy' constraint creation that includes task information directly. By using task related information and muscle information sources and then sparsification, the methods construct task relevant network communities among muscles, together with task redundant communities, and task irrelevant communities. This process of creating network communities may then constrain and help to guide subsequent synergy identification using the authors published sNM3F algorithm to detect spatial and temporal synergies. The revised paper is now much clearer and examples are helpful in various ways.

The impact of the information theoretic constraints developed as network communities on subsequent synergy separation are posited to be benign and to improve separation and identification of synergies over other methods (e.g., NNMF). However, not fully addressed are the possible impacts of the methods on the resulting compositionality and its links with physiological bases: the possibility remains that the methods here sometimes will instead lead to modules that represent more descriptive ML frameworks for task description, and resulting 'synergies' that may not support physiological work easily. Accordingly, there is a caveat for users of this framework. This is recognized and acknowledged by the authors in their rebuttal letters responding to prior reviews. It will remain for other work to explore this issue, likely through testing on detailed high degree of freedom artificial neuromechanical models and tasks. This possible issue and caveat with the strategy proposed by the authors likely should be more fully acknowledged in the paper.

The approach of the methods seeks to identify task relevant coordinative couplings. This identification is a meta problem for more classical synergy analyses. Classical/prior analyses seek compositional elements stable across tasks. These elements may then be explored in causal experiments and in generative simulations of coupling and control strategies. However, task-based understanding of synergy roles and functional uses as captured in the proposed methods are significant, and the field is clearly likely to be aided by methods in this study.

Information based separation has been used in muscle synergy analyses previously, by using infomax ICA, to discover physiological primitives. Though linear mixing of sources is assumed in ICA, minimized mutual information among source (synergy) drives is the basis of the separation and can detect low variance synergy contributions (e.g., see Yang, Logan, Giszter, 2019). In the work in the current paper, instead, mutual information approaches are used to cluster muscles and task features into network communities preceding the SNM3F algorithm use for separation, rather than using minimized information in the separation process directly. This contrast of an accretive or agglomerative mutual information strategy in the paper here, which is used to cluster into networks, versus a minimizing mutual information source separation used in infomax ICA epitomizes a key difference in approach. Indeed, physiological causal testing of synergy ideas is neglected in the literature reviews presented in the paper. Although these are only in animal work (e.g., Hart and Giszter, 2010; Takei and Seki, 2017), the clear connection of muscle synergy analysis choices to physiology is important, and eventually these issues need to be better managed and understood in relation to the new methods proposed here, even if not in this paper. Analyses of synergies using the methods the paper has proposed will likely be very much dependent on the number and quality of task variables included and how these are chosen, and the impacts of these on the ensuing sparsification and network communities used prior to SNM3F has already been noted. The authors acknowledge this in their responses. It would be useful in the future to explore the approach described with a range of simulated data to better understand the caveats, and optimizations for best practices in applications of this approach.

A key component of the authors' arguments here is their 'emergentist' view presented in the work, but perhaps not made fully explicit. Through the reductionist lens, which was used in the other physiological work noted above, muscle groupings are the units (primitives or 'building blocks' with informational separations) of coordinated movement and thus the space of these intermuscular unit interactions and controls is of particular interest for understanding movement construction and underlying physiology. This may allow representation of a hierarchy or heterarchy of neural control elements with clear physiological bases at spinal, brainstem and cortical levels. On the other hand, the emergentist view utilized by the authors here suggests that muscle groupings emerge from interactions between many constituent parts in a more freeform fashion with potentially larger task synergy assemblies (also quantified here using information tools). Information methods are applied differently using the two different lenses. The emergentist lens may potentially obscure fundamental neural controls and make them harder to explore in the descriptions resulting. Nonetheless, the different approaches to muscle synergy research, seeking different sorts of explanation and description of 'synergy', can be complementary and beneficial for the field overall going forward, so long as the caveats and concerns noted here are employed by readers in the interpretation of this new method.

---

## [Referee Report · Reviewer #3 (Public Review)]

In this study, the authors developed and tested a novel framework for extracting muscle synergies. The approach aims at removing some limitations and constrains typical of previous approaches used in the field. In particular, the authors propose a mathematical formulation that removes constrains of linearity and couple the synergies to their motor outcome, supporting the concept of functional synergies and distinguishing the task-related performance related to each synergy. While some concepts behind this work were already introduced in recent work in the field, the methodology provided here encapsulates all these features in an original formulation providing a step forward with respect to the currently available algorithms. The authors also successfully demonstrated the applicability of their method to previously available datasets of multi-joint movements.

Preliminary results positively support the scientific soundness of the presented approach and its potential. The added values of the method should be documented more in future work to understand how the presented formulation relates to previous approaches and what novel insights can be achieved in practical scenarios and confirm/exploit the potential of the theoretical findings.

---

## [Author Response]

The following is the authors’ response to the previous reviews.

**Public Reviews:**

**Reviewer #1 (Public Review):**
The proposed study provides an innovative framework for the identification of muscle synergies taking into account their task relevance. State-of-the-art techniques for extracting muscle interactions use unsupervised machine-learning algorithms applied to the envelopes of the electromyographic signals without taking into account the information related to the task being performed. In this work, the authors suggest including the task parameters in extracting muscle synergies using a network information framework previously proposed. This allows the identification of muscle interactions that are relevant, irrelevant, or redundant to the parameters of the task executed.The proposed framework is a powerful tool to understand and identify muscle interactions for specific task parameters and it may be used to improve man-machine interfaces for the control of prostheses and robotic exoskeletons.With respect to the network information framework recently published, this work added an important part to estimate the relevance of specific muscle interactions to the parameters of the task executed. However, the authors should better explain what is the added value of this contribution with respect to the previous one, also in terms of computational methods.It is not clear how the well-known phenomenon of cross-talk during the recording of electromyographic muscle activity may affect the performance of the proposed technique and how it may bias the overall outcomes of the framework.

We thank reviewer 1 for their useful commentary on this manuscript.

**Reviewer #2 (Public Review):**
This paper is an attempt to extend or augment muscle synergy and motor primitive ideas with task measures. The authors idea is to use information metrics (mutual information, co-information) in 'synergy' constraint creation that includes task information directly. By using task related information and muscle information sources and then sparsification, the methods construct task relevant network communities among muscles, together with task redundant communities, and task irrelevant communities. This process of creating network communities may then constrain and help to guide subsequent synergy identification using the authors published sNM3F algorithm to detect spatial and temporal synergies.The revised paper is much clearer and examples are helpful in various ways. However, figure 2 as presented does not convincingly show why task muscle mutual information helps in separating synergies, though it is helpful in defining the various network communities used in the toy example.The impact of the information theoretic constraints developed as network communities on subsequent synergy separation are posited to be benign and to improve over other methods (e.g., NNMF). However, not fully addressed are the possible impacts of the methods on compositionality links with physiological bases, and the possibility remains of the methods sometimes instead leading to modules that represent more descriptive ML frameworks that may not support physiological work easily. Accordingly, there is a caveat. This is recognized and acknowledged by the authors in their rebuttal of the prior review. It will remain for other work to explore this issue, likely through testing on detailed high degree of freedom artificial neuromechanical models and tasks. This possible issue with the strategy here likely needs to be fully acknowledged in the paper.The approach of the methods seeks to identify task relevant coordinative couplings. This is a meta problem for more classical synergy analyses. Classical analyses seek compositional elements stable across tasks. These elements may then be explored in causal experiments and generative simulations of coupling and control strategies. However, task-based understanding of synergy roles and functional uses is significant and is clearly likely to be aided by methods in this study.Information based separation has been used in muscle synergy analyses using infomax ICA, which is information based at core. Though linear mixing of sources is assumed in ICA, minimized mutual information among source (synergy) drives is the basis of the separation and detects low variance synergy contributions (e.g., see Yang, Logan, Giszter, 2019). In the work in this paper, instead, mutual information approaches are used to cluster muscles and task features into network communities preceding the SNM3F algorithm use for separation, rather than using minimized information in separation. This contrast of an accretive or agglomerative mutual information strategy here used to cluster into networks, versus a minimizing mutual information source separation used in infomax ICA epitomizes a key difference in approach here.Physiological causal testing of synergy ideas is neglected in the literature reviews in the paper. Although these are only in animal work (Hart and Giszter, 2010; Takei and Seki, 2017), the clear connection of muscle synergy analysis choices to physiology is important, and eventually these issues need to be better managed and understood in relation to the new methods proposed here, even if not in this paper.Analyses of synergies using the methods the paper has proposed will likely be very much dependent on the number and quality of task variables included and how these are managed, and the impacts of these on the ensuing sparsification and network communities used prior to SNM3F. The authors acknowledge this in their response. This caveat should likely be made very explicit in the paper.It would be useful in the future to explore the approach described with a range of simulated data to better understand the caveats, and optimizations for best practices in this approach.

A key component of the reviewers’ arguments here is their reductionist view of muscle synergies vs the emergentist view presented in our work here. In the reductionist lens, muscle groupings are the units (‘building blocks’) of coordinated movement and thus the space of intermuscular interactions is of particular interest for understanding movement construction. On the other hand, the emergentist view suggests that muscle groupings emerge from interactions between constituent parts (as quantified here using information theory, synergistic information is the information found when both activities are observed together). This is in line with recent work in the field showing modular control at the intramuscular level, exemplifying a scale-free phenomena. Nonetheless, we consider these approaches to muscle synergy research as complementary and beneficial for the field overall going forward.

**Reviewer #3 (Public Review):**
In this study, the authors developed and tested a novel framework for extracting muscle synergies. The approach aims at removing some limitations and constraints typical of previous approaches used in the field. In particular, the authors propose a mathematical formulation that removes constraints of linearity and couples the synergies to their motor outcome, supporting the concept of functional synergies and distinguishing the task-related performance related to each synergy. While some concepts behind this work were already introduced in recent work in the field, the methodology provided here encapsulates all these features in an original formulation providing a step forward with respect to the currently available algorithms. The authors also successfully demonstrated the applicability of their method to previously available datasets of multi-joint movements.Preliminary results positively support the scientific soundness of the presented approach and its potential. The added values of the method should be documented more in future work to understand how the presented formulation relates to previous approaches and what novel insights can be achieved in practical scenarios and confirm/exploit the potential of the theoretical findings.In their revision, the authors have implemented major revisions and improved their paper. The work was already of good quality and now it has improved further. The authors were able to successfully:improve the clarity of the writing (e.g.: better explaining the rationale and the aims of the paper);extend the clarification of some of the key novel concepts introduced in their work, like the redundant synergies;show a scenario in which their approach might be useful for increasing the understanding of motor control in patients with respect to traditional algorithms such as NMF. In particular, their example illustrates why considering the task space is a fundamental step forward when extracting muscle synergies, improving the practical and physiological interpretation of the results.

We thank reviewer 3 for their constructive commentary on this manuscript.

**Recommendations for the authors:**

**Reviewer #1 (Recommendations For The Authors):**
Figure 3 should report the distances between reaching points in panel A and the actual length distances of the walking paths in panel C.

The caption of fig.3 concerning the experimental setup of the datasets analysed has been updated with the following for dataset 1: “(A) Dataset 1 consisted of participants executing table-top point-to-point reaching movements (40cm distance from starting point P0) across four targets in forward (P1-P4) and backwards (P5-P8) directions at both fast and slow speeds (40 repetitions per task) [25]. The muscles recorded included the finger extensors (FE), brachioradialis (BR), biceps brachii (BI), medial-triceps (TM), lateral-triceps (TL), anterior deltoid (AD), posterior deltoid (PD), pectoralis major (PE), latissimus dorsi (LD) of the right, reaching arm.”. For dataset 3, to the best of the authors knowledge, this information was not given in the original paper.

Figure 4, what is the unit of the data shown?

The unit of bits is now mentioned in the toy example figure caption and in the caption of fig.5

Figure 4, the characteristics of the interactions are not fully clear, and the graphical representation should be improved.

We have made steps to improve the clarity of the figures presented.

For dataset 3, τ was the movement kinematics, but it is not specified how the task parameters were formulated. Did the authors use the data from all 32 kinematic markers, 4 IMUs, and force plates? If yes, it should be specified why all these signals were used. For sure, there will be signals included that are not relevant to the specific task. Did the authors select specific signals based on their relevance to the task (e.g., ankle kinematics)?

We have now clarified this in the text as follows: “For datasets 1 and 2, we determine the MI between vectors with respect to several discrete task parameters representing specific task attributes (e.g. reaching direction, speed etc.), while for dataset 3 we determined the task-relevant and -irrelevant muscles couplings in an unassuming way by quantifying them with respect to all available kinematic, dynamic and inertial motion unit (IMU) features.”

How did the authors endure that crosstalk did not affect their analysis, particularly between, e.g., finger extensors and brachioradialis and posterior deltoid and anterior deltoid (dataset 1)?

We have addressed this point in the previous round of reviews and made an explicit statement regarding cross-talk in the discussion section: “Although distinguishing task-irrelevant muscle couplings may capture artifacts such as EMG crosstalk, our results convey several physiological objectives of muscles including gross motor functions [66], the maintenance of internal joint mechanics and reciprocal inhibition of contralateral limbs [19,51].”

It would be informative to add some examples of not trivial/obvious task-related synergistic muscle combinations that have been extracted in the three datasets. Most of the examples reported in the manuscript are well-known biomechanically and quite intuitive, so they do not improve our understanding of synergistic muscle control in humans.

Our framework improves our understanding of synergistic motor control by enabling the formal quantification of synergistic muscle interactions, a capability not present among current approaches. Regarding the implications of this advance in terms of concrete examples, we have further clarified our examples presented in the results section, for example:

“Across datasets, many the muscle networks could be characterised by the transmission of complementary task information between functionally specialised muscle groups, many of which identified among the task-redundant representations (Fig.9-10 and Supp. Fig.2). The most obvious example of this is the S3 synergist muscle network of dataset 2 (Fig.11), which captures the complementary interaction between task-redundant submodules identified previously (S3 (Fig.9)).”

The description shows how our framework can extract the cross-module interactions that align with the higher-level objectives of the system, here the synergistic connectivity between the upper and lower body modules. Current approaches can only capture redundant and task-irrelevant interactions. Thus our framework provides additional insight into movement control.

The number of participations in dataset 2 is very limited and should be increased.We appreciate the reviewer's comment and would like to point out that for dataset 2 our aim was to increase the number of muscles (30), tasks (72) and trials for each task (30) which produced a very large dataset for each participant. This came at the expense of low number of participants, however all our statistical analyses here can be performed at the single-participant level. Furthermore, dataset 3 includes 25 participants and it enables us to demonstrate the reliability of the findings across participants.
**Reviewer #2 (Recommendations For The Authors):**
I believe it is important in the future to explore the approach proposed with a range of simulation data and neuromechanical models, to explore the issues I have raised and that you have acknowledged, though I agree it is likely out of scope for the paper here.

We agree with the reviewer that this would be valuable future work and indeed plan to do this in our future research.

The Github code for this paper should likely include the various data sets used in the paper and figures, appropriately anonymized, in order to allow the data to be explored and analyses replicated and package demonstrated to be exercised fully by a new user.

We thank the reviewer for this suggestion. Dataset3 is already available online at https://doi.org/10.1016/j.jbiomech.2021.110320. We will also make the other 2 datasets publicly available on our lab website very soon. Until then, as stated in the manuscript, we will make them available to anyone upon reasonable request.

**Reviewer #3 (Recommendations For The Authors):**
I have the following open points to suggest to the authors:First, I recommend improving the quality of the figures: in the pdf version I downloaded, some writings are impossible to read.

We fully agree with the reviewer and note that in the pdf version of the paper, the figures are a lot worse than in the submitted word document submitted. Nevertheless, we will make further improvements on the figures as requested.

Even though the manuscript has improved, I still feel that some points were not addressed or were only partially addressed. In particular:The proposed comparison with NMF helps understanding why incorporating the task space is useful (and I fully agree with the authors about this point as the main reason to propose their contribution). However, the comparison does not help the reader to understand whether the synergies incorporating the task space are biased by the introduction of the task variables.This question can be also reformulated as: are muscle synergies modified when task space variables are incorporated? Is the "weight" on task coefficients affecting the composition of muscle synergies? If so, the added interpretational power is achieved at the cost of losing the information regarding the neural substrate of synergies? I understand this point is not immediate to show, but it would increase the quality of the work.Reference to previous approaches that aimed at including task variables into synergy extraction are still missing in the paper. Even though it is not required to provide quantitative comparisons with other available approaches, there are at most 2-3 available algorithms in the literature (kinematics-EMG; force-EMG), that should not be neglected in this work. What did previous approaches achieve? What was improved with this approach? What was not improved?Previous attempts of extracting synergies with non-linear approaches could also be described more.

In the latest version of the manuscript, we have referenced both the mixed NMF and autoencoders based algorithms. In both the introduction and discussion section of the manuscript, we also specify that our framework quantifies and decomposes muscle interactions in a novel way that cannot be done by other current approaches. In the results section we use examples from 3 different datasets to make this point clear, providing intuition on the use cases of our framework.